# The Ten Dietary Commandments for Patients with Irritable Bowel Syndrome: A Narrative Review with Pragmatic Indications

**DOI:** 10.3390/nu17152496

**Published:** 2025-07-30

**Authors:** Nicola Siragusa, Gloria Baldassari, Lorenzo Ferrario, Laura Passera, Beatrice Rota, Francesco Pavan, Fabrizio Santagata, Mario Capasso, Claudio Londoni, Guido Manfredi, Danilo Consalvo, Giovanni Lasagni, Luca Pozzi, Vincenza Lombardo, Federica Mascaretti, Alice Scricciolo, Leda Roncoroni, Luca Elli, Maurizio Vecchi, Andrea Costantino

**Affiliations:** 1Department of Pathophysiology and Transplantation, University of Milan, 20122 Milan, Italy; nicola.siragusa@unimi.it (N.S.); francesco.pavan@unimi.it (F.P.); fabrizio.santagata@unimi.it (F.S.); maurizio.vecchi@unimi.it (M.V.); 2Department of Biosciences, University of Milan, 20133 Milan, Italy; gloria.baldassari@studenti.unimi.it (G.B.); lorenzo.ferrario6@studenti.unimi.it (L.F.); laura.passera2@studenti.unimi.it (L.P.); 3Gastroenterology and Endoscopy Unit, Fondazione IRCCS Ca’ Granda Ospedale Maggiore Policlinico, 20122 Milan, Italy; beatrice.rota@policlinico.mi.it (B.R.); luca.pozzi@policlinico.mi.it (L.P.); vincenza.lombardo@policlinico.mi.it (V.L.); federica.mascaretti@policlinico.mi.it (F.M.); alice.scricciolo@policlinico.mi.it (A.S.); leda.roncoroni@unimi.it (L.R.); luca.elli@policlinico.mi.it (L.E.); 4Gastroenterology and Endoscopy Unit, ASST Ospedale Maggiore, 26013 Crema, Italy; mario.capa05@gmail.com (M.C.); c.londoni@asst-crema.it (C.L.); guido.manfredi@asst-crema.it (G.M.); 5Department of Gastroenterology and Digestive Endoscopy, AORN “Antonio Cardarelli”, 80131 Naples, Italy; dr.consalvo@gmail.com; 6Unit of Immunology, Rheumatology, Allergy and Rare Diseases (UniRAR), IRCCS San Raffaele Hospital, 20132 Milan, Italy; giovanni.lasagni@hsr.it; 7Department of Biomedical, Surgical and Dental Sciences, University of Milan, 20122 Milan, Italy

**Keywords:** irritable bowel syndrome, DGBI, diet, personalized dietary guidance

## Abstract

Irritable bowel syndrome (IBS) is a gut–brain axis chronic disorder, characterized by recurrent abdominal pain and altered bowel habits in the absence of organic pathology. Nutrition plays a central role in symptom management, yet no single dietary strategy has demonstrated universal effectiveness. This narrative review critically evaluates current nutritional approaches to IBS. The low-Fermentable Oligo-, Di-, Mono-saccharides and Polyols (FODMAP) diet is the most extensively studied and provides short-term symptom relief, but its long-term effects on microbiota diversity remain concerning. The Mediterranean diet, due to its anti-inflammatory and prebiotic properties, offers a sustainable, microbiota-friendly option; however, it has specific limitations in the context of IBS, particularly due to the adverse effects of certain FODMAP-rich foods. A gluten-free diet may benefit individuals with suspected non-celiac gluten sensitivity, although improvements are often attributed to fructan restriction and placebo and nocebo effects. Lactose-free diets are effective in patients with documented lactose intolerance, while a high-soluble-fiber diet is beneficial for constipation-predominant IBS. IgG-based elimination diets are emerging but remain controversial and require further validation. In this review, we present the 10 dietary commandments for IBS, pragmatic and easily retained recommendations. It advocates a personalized, flexible, and multidisciplinary management approach, avoiding rigidity and standardized protocols, with the aim of optimizing adherence, symptom mitigation, and health-related quality of life. Future research should aim to evaluate, in real-world clinical settings, the impact and applicability of the 10 dietary commandments for IBS in terms of symptom improvement and quality of life

## 1. Introduction

Irritable bowel syndrome (IBS) remains one of the most common gastrointestinal disorders for physicians in both primary and secondary care [1], with a prevalence ranging from 3.8% to 12% in the general population. This variability is influenced by diagnostic criteria and geographic location, with a higher incidence observed in women [2,3,4]. Specifically, the global prevalence of IBS has been estimated at 4.1% based on the Rome IV diagnostic criteria (5% in Italy), compared to 9.2% using the Rome III criteria [5,6]. These estimates were derived from a systematic review and meta-analysis of studies conducted across 38 countries, involving a total of approximately 400,000 participants [7].

In recent years, significant advances have been made in understanding the complex pathophysiology of IBS, leading to its reclassification under the 2016 Rome IV criteria as a disorder of gut–brain interaction (DGBI), rather than as a functional gastrointestinal disorder [1,5].

DGBIs represent one of the leading reasons for gastroenterological evaluation [2] and are associated with significant global healthcare costs [3,4]. Patients with IBS experience chronic or recurrent abdominal pain (occurring at least one day per week over the past three months, with onset at least six months before the diagnosis) associated with defecation and/or altered bowel habits [5]. These features should be present in the absence of positive findings on simple recommended tests used to exclude organic diseases (e.g., complete blood count, C-reactive protein, fecal calprotectin, and celiac serology) and alarm symptoms. Patients are classified into subgroups based on their predominant bowel pattern as follows: (1) IBS with diarrhea (IBS-D), (2) IBS with constipation (IBS-C), (3) IBS with mixed bowel habits (IBS-M), or (4) unclassified IBS (IBS-U). The presence of abdominal pain is the key feature that distinguishes IBS from other functional bowel disorders, which include functional constipation, functional diarrhea, and functional abdominal bloating or distension [8]. However, there is a certain degree of overlap and fluctuation between IBS and these other DGBI [9,10]. The exact cause of IBS is unknown, but it is believed to result from a combination of factors, including dysfunction of the enteric nervous system (ENS), visceral hypersensitivity, psychological factors such as stress, anxiety, and depression, dietary influences, and alterations in the gut microbiota [5]. Indeed, IBS is a complex condition involving not only the gastrointestinal tract but also the central nervous system (CNS) and the ENS. Numerous treatment options are available to help manage symptoms and improve patients’ quality of life. The British Society of Gastroenterology guidelines for the management of IBS [5] recommend an integrated approach, which may include a combination of targeted pharmacological, nutritional, and psychological therapies. Moreover, it is important to consider the beneficial effects of physical activity and exercise training on the symptoms of IBS [11]. Randomized controlled trials (RCTs) have shown that exercise can help improve IBS symptoms [12,13,14]. Specifically, three studies [12,14] reported significant improvements in overall IBS symptoms, as measured by the IBS-SSS (IBS Symptom Severity Score). Another study [14] found that exercise significantly improved only constipation symptoms in IBS patients. The amount and type of physical activity used in these studies generally aligned with the World Health Organization (WHO) guidelines.

Diet plays a fundamental role in human life, not only from a nutritional perspective but also in social and cultural terms [15]. Originally, the term ‘diet’, derived from the Greek *díaita*, referred to an overall lifestyle rather than simply a set of dietary restrictions. Today, diet is seen not only as nutrition but also as culture, identity, and sociality. Indeed, each population develops its own dietary model over time, shaped by history, geography, religion, and social relationships [16]. Therefore, food becomes a symbol, a ritual, and a language that unites people and strengthens the sense of belonging to a community [17]. Mealtime often represents an opportunity for sharing and dialogue, reinforcing family and social bonds [18,19].

With globalization, the introduction of fast food, and changes associated with modernity, there is an increasing need to make ethical choices inspired, for example, by vegetarian or vegan dietary principles. In this context, diet becomes an educational issue: it is essential to learn how to eat a balanced diet, read food labels, and make informed choices, not only for personal well-being but also for the environment [20,21]. In recent years, food production has developed along two distinct but parallel paths. On one hand, there is a focus on quality, closely linked to the regions of origin, local traditions, and the cultural heritage of communities. On the other hand, a production model based on quantity has emerged, moving away from taboos deeply rooted in our culture [22], where food results from fragmented processes that include production, processing, and distribution phases located in different parts of the world [23,24].

In this second approach, food is able to nourish the body but loses the capacity to feed the mind and culture of the consumer. Eating, in fact, is not a purely mechanical or trivial act: it is a gesture that impacts physical and mental health, generates pleasure, strengthens social bonds, and fulfills the human need for knowledge, discovery, and connection with the world [25].

Dietary patterns have evolved over time in response to scientific discoveries and individual health needs [26]. In contemporary times, there has been growing interest in personalized dietary regimens, often adopted for clinical, functional, or preventive purposes [26]. Among these, certain diets have gained popularity due to their effectiveness in managing specific disorders, improving digestive well-being, or preventing adverse reactions [27,28].

This narrative review aims to comprehensively examine the primary dietary strategies employed in patients with irritable bowel syndrome (IBS), integrating the current evidence base with clinical expertise to propose a practical model for nutritional management. A systematic literature search was conducted across PubMed, Scopus, Web of Science, Google Scholar, and the Cochrane Library, considering articles published in English up to July 2025. Boolean operators were applied to combine the following search terms: (“irritable bowel syndrome” OR “IBS” OR “disorders of gut–brain interaction”) AND (“diet” OR “nutrition” OR “low-Fermentable Oligo-, Di-, Mono-saccharides and Polyols (FODMAP)” OR “gluten-free” OR “lactose-free” OR “fiber” OR “elimination diet” OR “Mediterranean diet”).

Study selection was performed independently by four reviewers (N.S., G.B., L.F., and L.P.), who screened titles, abstracts, and, when necessary, full texts. A total of 247 articles were initially identified. Following the removal of duplicates and the application of predefined inclusion criteria, 188 articles were included in the final analysis. Inclusion criteria encompassed English-language, peer-reviewed publications focusing on dietary interventions in IBS, with particular emphasis on randomized controlled trials (RCTs), meta-analyses, systematic reviews, and clinical guidelines. Non-relevant studies, non-peer-reviewed articles, and publications in languages other than English were excluded. Selected clinically relevant case reports were incorporated when deemed illustrative or contextually informative.

Discrepancies among reviewers during study selection were resolved through discussion and consensus. Although this review does not strictly adhere to a formal PRISMA protocol, a systematic and transparent methodology was employed for literature retrieval and appraisal to ensure the reliability and reproducibility of the reported evidence. Dietary management of IBS must be individualized, flexible, and multidimensional. The “Ten Commandments” offer an innovative and pragmatic approach that shifts the focus away from standardized food prescriptions towards the identification and correction of ineffective or harmful behaviors, such as self-diagnosis, social isolation, aesthetic obsession, and the adoption of “one-size-fits-all” diets. This patient-centered, evidence-based model enhances therapeutic adherence, empowers individuals, and contributes to a sustainable improvement in quality of life.

Below is an overview of some of the main diets currently used for therapeutic or functional purposes:**Low-FODMAP Diet**: Specifically indicated for IBS, it reduces the intake of short-chain fermentable carbohydrates, which can lead to intestinal fermentation and symptoms such as bloating, abdominal pain, and altered bowel habits.**Mediterranean Diet**: Rich in antioxidants, fiber, and healthy fats, it may help reduce low-grade inflammation and modulate the composition of the gut microbiota, contributing to symptom control in IBS.**IgG-Based Elimination Diet**: Sometimes used in IBS to identify potential trigger foods, this approach may help alleviate symptoms, although scientific evidence supporting its efficacy remains debated.**Diets with Soluble or Insoluble Fiber**: Soluble fibers (e.g., psyllium) are often effective in improving bowel regularity and reducing pain and bloating in IBS patients. Insoluble fibers should be used with caution, as they may sometimes exacerbate symptoms.**Lactose-Free Diet**: Often beneficial in IBS patients with associated lactose intolerance; eliminating lactose can help reduce gas, cramps, and diarrhea.**Gluten-Free Diet**: Some IBS patients report symptom improvement when eliminating gluten, even in the absence of celiac disease; it may be useful in cases of suspected non-celiac gluten sensitivity.

Across these dietary models, the low-FODMAP diet emerged as the most effective for symptom relief, showing benefit in about 70% of IBS patients, especially for bloating and abdominal pain. However, long-term adherence may reduce microbial diversity, particularly Bifidobacteria, thus requiring professional supervision. The Mediterranean diet, despite its richness in FODMAP, is sustainable and microbiota-friendly, with anti-inflammatory and metabolic advantages if properly adapted. Gluten- and lactose-free diets can provide benefit in select subgroups, though evidence points to fructan and disaccharide intolerance rather than gluten or lactose per se. Soluble fiber supplementation is useful, especially in IBS-C, while insoluble fiber may worsen symptoms in IBS-D. IgG-based elimination remains controversial, requiring cautious and individualized application.

In conclusion, the dietary management of IBS must be dynamic and tailored to the individual. There is no universal solution; nutritional interventions should be guided by the patient’s phenotype, individual tolerance, lifestyle, and microbiota response. Rigid self-prescription may worsen symptoms and impair quality of life. Conversely, a flexible, interdisciplinary approach (integrating nutrition, clinical care, and psychological support) represents the most effective strategy. The goal is not dietary perfection, but a sustainable, personalized pathway for each patient.

## 2. Dietary Approaches and Their Implications for IBS (Table 3)

### 2.1. The Low-FODMAP Diet

Normal intestinal motility in patients with IBS may be perceived as dysfunctional, leading to a condition known as visceral hypersensitivity. This condition is often exacerbated by the ingestion of foods with high osmolarity and poor absorption, such as FODMAP [29].

These compounds are defined as short-chain carbohydrates that are poorly absorbed and can be fermented by the gut microbiota, resulting in gas production, intestinal lumen distension, and the synthesis of neuroactive mediators that amplify nociceptive signaling, thereby contributing to the exacerbation of symptoms [30,31].

An additional pathogenetic mechanism involved in IBS concerns increased intestinal barrier permeability. This may facilitate the translocation of antigens through the epithelial tight junctions, leading to mast cell activation and triggering an inflammatory cascade that contributes to symptom occurrence [32]. Moreover, alterations in the gut microbiota may enhance the fermentation of FODMAP [33], modifying the body’s response and the disease subtype, while intensifying luminal distension and the activation of visceral nociceptive responses [34]. It has also been hypothesized that certain nutrients or bioactive food compounds can activate specific receptors on enterocytes, modulating not only absorption but also the communication between the gut, nervous system, and immune system, thereby triggering what is commonly referred to as the gut–brain axis [35]. These signals can influence motility, intestinal sensitivity, and mucosal permeability, further contributing to symptom genesis. Currently, the low-FODMAP diet represents the nutritional intervention with the strongest scientific evidence for managing IBS and is one of the most studied and widely applied in clinical practice [36].

This type of diet is based on reducing the intake of FODMAPs, which are short-chain fermentable carbohydrates that are incompletely absorbed in the small intestine and subsequently fermented by the microbiota in the colon [37]. This process generates gas and draws water into the intestinal lumen, contributing to the onset of the main symptoms of IBS, including abdominal pain, bloating, flatulence, and diarrhea. The low-FODMAP diet helps control symptoms, reduce abdominal pain and bloating, and improve quality of life in these patients [38].

The protocol consists of three distinct phases [38,39]. (1) Elimination phase (2–8 weeks): During this period, high-FODMAP foods are avoided and replaced with low-FODMAP alternatives from the same food group. If no clinical improvement is observed within this timeframe, alternative therapeutic strategies are recommended. (2) Gradual reintroduction phase (rechallenge): High-FODMAP foods are reintroduced individually every 2–3 days to identify specific triggers and assess individual tolerance. (3) Maintenance phase: A personalized diet is established that excludes only poorly tolerated foods while reintroducing well-tolerated ones, aiming to ensure the diet remains as varied and balanced as possible.

This diet is particularly useful because it allows for extensive personalization, adapting to the patient’s preferences and nutritional needs, thereby significantly improving quality of life. However, the second and third phases are often difficult to implement without the support of a dietitian, nutritionist, or a trained specialist in managing these functional disorders.

In a recent meta-analysis, the low-FODMAP diet demonstrated clinically significant effects in reducing gastrointestinal symptoms, such as bloating and abdominal distension, in adult patients with IBS of any subtype. Specifically, it ranked first in improving bloating compared to alternative diets, including habitual diets, high-FODMAP diets, and generic dietary approaches [40].

A clinical study compared the effects of three diets (low-FODMAP, gluten-free, and traditional balanced) in patients with IBS. The low-FODMAP diet proved superior in reducing abdominal bloating compared to the other two [41]. However, some studies have raised concerns about the long-term sustainability of the low-FODMAP diet. Prolonged adherence to this diet may negatively affect the diversity and composition of the gut microbiota, reducing the presence of bacteria beneficial to human health [42].

In addition, the low-FODMAP diet reduces the intake of substances such as fructo-oligosaccharides (FOSs) and galacto-oligosaccharides (GOSs), which act as natural prebiotics. This restriction can decrease *Bifidobacteria*, beneficial bacteria for gut health, without significantly altering overall microbial diversity. This underscores the importance of limiting the restrictive phase and gradually reintroducing FODMAPs [42].

Furthermore, it is hypothesized that certain dietary stimuli may activate abnormal response mechanisms in the intestine through interaction with specific receptors present on the enterocyte membrane [43]. This suggests that the diet’s efficacy may also be due to direct modulation of intestinal sensitivity, in addition to the reduction in fermentation.

Moreover, a low-FODMAP diet can be challenging for patients to follow. One of the main difficulties involves ensuring adequate intake of dietary fiber and calcium.

Therefore, although the low-FODMAP diet is an effective short-term intervention, it should be considered part of an integrated nutritional strategy, tailored to the individual characteristics of the patient and regularly monitored to ensure lasting benefits without compromising gut health. Dietary counseling by specialized personnel is certainly key to the success of this rather complex nutritional approach. Once a good symptomatic response is achieved, patients can be supported in gradually reintroducing all the foods they can tolerate, with the ultimate goal of transitioning to a personalized diet based on symptoms experienced during food reintroduction. If no symptom improvement occurs after 4–8 weeks, the intervention should be discontinued, and alternative therapeutic options considered [39].

Table 1 shows a classification of foods based on their FODMAP content, divided into three main categories: allowed foods, foods to be consumed in moderation, and foods to avoid. This distinction is useful for guiding dietary choices in patients with IBS or functional gastrointestinal disorders, supporting a more effective and personalized nutritional approach.

### 2.2. The Mediterranean Diet

The term “Mediterranean diet” (MD) was first coined in the 1960s by Ancel Keys, an American biologist, physiologist, and epidemiologist, following the results of the Seven Countries Study. This epidemiological study showed evidence of a lower mortality rate and incidence of cardiovascular diseases and cancer among populations in the Mediterranean area, such as Italy and Greece, compared to other populations [44].

MD is a generalization of the traditional dietary pattern of the inhabitants of the Mediterranean region. It is characterized by a wide variety and abundance of whole grains, non-starchy vegetables, legumes, nuts, and seeds, which were staple foods for both men and women in traditional cultures. At the same time, meat, fish, milk, cheese, and eggs were consumed only a few times a week and were considered high-value foods (Table 2) [45,46].

Numerous studies have emphasized that the combination of MD and a healthy lifestyle is beneficial for individual health. These benefits have been particularly observed in relation to age-related non-communicable diseases (NCDs), such as metabolic and cardiovascular diseases, neurodegenerative disorders, cancer, depression, respiratory illnesses, and bone frailty. In addition to all these advantages, the MD represents a lifestyle that promotes healthy eating while maintaining a strong connection with both environmental and economic sustainability [47]. Low adherence to MD is associated with a higher prevalence of IBS. However, certain foods typical of the MD may exacerbate symptoms in patients suffering from this condition [48].

A clinical study conducted in Australia involved 59 individuals (29 following an MD and 30 controls) who suffered from moderate to severe IBS, with symptoms occurring at least twice a week and associated psychological distress. The results showed that the MD was a diet easy to implement in daily life, with high adherence, and led to significant improvements compared to the control group in gastrointestinal symptoms (83% in the MD group vs. 37% in controls) and depressive symptoms (62% in the MD group vs. 23% in controls) [49]. At the same time, it is important to consider that foods that are an integral part of the MD are often associated with the onset of IBS symptoms, as they contain high levels of FODMAP (such as fruits, vegetables, and legumes). For example, a high fiber intake may worsen symptoms in patients with IBS-D, but may be beneficial for those with IBS-C [50]. These issues often lead patients with IBS to eliminate or reduce the intake of specific food groups they believe are responsible for their discomfort. However, if carried out without the clinical guidance of a healthcare professional, this practice may potentially lead to nutritional deficiencies and negatively impact the individual’s overall health [51].

In reality, the diet itself should not be seen as directly responsible, as it is modifiable. For this reason, it has been shown that the application of the MD should be personalized, with the support of a nutrition professional, and by taking into account the worsening of symptoms in relation to each individual [48].

Another important aspect to consider is that an increasing portion of the population follows vegan or vegetarian diets, which makes it more challenging to find a dietary approach that helps reduce IBS-related symptoms. These diets must ensure a high intake of legumes as a primary protein source in place of animal-based foods like meat and fish, as well as fiber from fruits and vegetables. However, some of these foods are high in FODMAPs, and a generalized elimination could lead to serious nutritional deficiencies. Once again, the involvement of a qualified professional is essential to properly manage all these factors [52,53].

As is well known, adherence to a specific diet is associated with characteristic alterations in the gut microbiota. In the case of the MD, a randomized controlled trial (RCT) involving participants from five European countries (the United Kingdom, France, the Netherlands, Italy, and Poland) demonstrated a positive association with several markers of greater physical resilience, improved health status, and reduced vulnerability, particularly in older adults or individuals with chronic conditions. Moreover, a negative interaction was observed with inflammatory markers such as C-reactive protein and interleukin-17 [54]. In addition, high adherence to a personalized MD has been associated with a low abundance of potentially harmful bacteria such as *Faecalitalea*, *Streptococcus*, and *Intestinibacter*, and a higher abundance of beneficial organisms like *Holdemanella* from the Phylum *Firmicutes*. Therefore, this could have a positive impact on the individual’s gut microbiota [48].

The MD is characterized by a high abundance of antioxidant and anti-inflammatory compounds. The fats consumed through the MD are mostly monounsaturated fatty acids (MUFAs) derived from extra virgin olive oil. There is also a high intake of omega-3 fatty acids from fish and plant sources, with a low omega-6 to omega-3 ratio (2:1–1:1 compared to 14:1 in other European countries). Additionally, the high consumption of fiber, foods containing antioxidant compounds, and polyphenols may lead to favorable anti-inflammatory effects at the intestinal level, where the low-FODMAP diet may not be effective.

The immunomodulatory capacity of this diet is important, in addition to its ability, as previously discussed, to increase the number of beneficial microorganisms for humans. Recently, the need to supplement the low-FODMAP diet with fibers and probiotics has been highlighted to, respectively, limit constipation and dysbiosis. Nevertheless, similar effects could be achieved through adequate consumption of allowed fruits and vegetables, which are rich in prebiotics and abundant in the MD [55].

Despite all these well-known positive aspects, adherence to the MD has been steadily declining in recent years due to various reasons related to social and cultural changes, as well as the increasing globalization of food habits [56,57]. Foremost among these is the rising cost of purchasing food, which represents a socioeconomic issue that is leading more and more inhabitants of the Mediterranean region to turn to the more affordable Western Diet [57].

Other reasons lie in individuals’ preference for foods that are pleasing to the senses and tasty, or in the limited variety and choice available, as revealed by a study on adherence to the MD conducted in five Mediterranean Basin countries (Greece, Italy, Morocco, Slovenia, and Tunisia). The same study shows that among the criteria less observed in food choices are origin and seasonality, as well as a restrictive selection mainly linked to aversions toward the taste or smell of certain foods.

Overall, since attitudes toward healthy eating were found to be the most significant predictor of adherence to the MD in these five countries, it is recommended to design targeted intervention strategies to provide information that increases consumer awareness about the health impact of balanced and healthier eating behaviors [58].

Finally, considering the distinctive characteristics of both the MD and the Low-FODMAP diet, it is indisputably known that the former has anti-inflammatory properties, while the latter is effective in alleviating IBS symptoms and is used as a first-line treatment. A recent RCT has highlighted the suitability of a new dietary protocol, the Mediterranean Low-FODMAP Diet, which could combine the benefits of both diets and reduce their limitations to promote patient well-being [59].

This once again demonstrates the need for a comprehensive, multi-faceted approach to treating this condition, guided by an expert professional. There is no single effective treatment; it must be personalized as much as possible according to the patient’s symptoms and adjusted consistently based on their improvement or worsening.

### 2.3. IgG Antibody-Based Food Elimination Diet

Patients suffering from IBS often believe that intolerance to specific foods is the main trigger for their symptoms. For this reason, they tend to eliminate these foods from their diet on their own. However, determining the presence of a true immunological response to these foods is very challenging: in the past, research focused on detecting a classic IgE-mediated allergic response, but with unsatisfactory results; in fact, it is rare for these conditions to involve an immediate immune response like in allergies.

This has led to the hypothesis that a more prolonged response over time could be mainly due to IgG antibodies, which have often been implicated in food hypersensitivity episodes. This theory is highly debated, and opinions are divided, primarily because food-related IgG antibodies can also be found in healthy individuals [60].

The complex immune response of our body also involves IgG antibodies. The lymphatic tissues of the gastrointestinal mucosa recognize as non-self the foods that cannot be completely digested due to the lack of specific enzymes. This leads to the formation of complexes with specific IgG antibodies, which initiate an immune response by binding to the corresponding antigens. Subsequently, macromolecular complexes are phagocytosed by monocytes, while micromolecular ones are eliminated through the kidneys [61]. Thus, IgG, especially IgG4, represents a normal adaptive immune response to dietary antigen exposure, contributing to the development of oral tolerance to food antigen [62].

Considering that high levels of IgG found in serum may indicate immunological activation and food hypersensitivity, an elimination diet based on these results could represent a personalized dietary approach that may lead to greater treatment effectiveness compared to a non-individualized diet, such as the low-FODMAP diet. It should be noted, however, that some FODMAP foods would also be avoided in an elimination diet [63].

Studies have demonstrated benefits in the treatment of IBS symptoms through an IgG-based elimination diet, possibly in combination with probiotic treatments [60,64].

Recent randomized evidence supports this approach: a sham-controlled trial demonstrated that an IBS-specific IgG ELISA-guided elimination diet significantly reduced symptom severity, particularly in patients with IBS-C and IBS-M subtypes. As dietary responses in IBS vary widely among individuals, tailoring interventions is essential [65].

However, the effect of food-specific IgG antibodies on human health remains controversial. IgG antibodies bind to their corresponding food antigens, and this process induces a mild inflammation in the body, which may lead to various systemic and pathological symptoms over the long term [66].

IgG antibodies can also be present in healthy individuals, although their concentrations remain lower than in pathological subjects. A study investigated antibody responses following the consumption of 14 foods typical of Chinese cuisine: higher IgG levels were found in women compared to men, as well as in older individuals, but the differences between healthy and ill individuals were not particularly pronounced [61].

Another study highlighted that serum concentrations of IgG antibodies increase in patients with IBS or functional gastrointestinal disorders in response to specific foods. However, no significant correlation has been found between symptom severity and elevated serum IgG antibodies in these patients [67].

That said, it is quite unlikely that the immune response triggered by food antigens is the sole explanation for postprandial symptoms in IBS. While part of the response is certainly activated by these antigens, symptoms may also be due to intestinal carbohydrate fermentation, gas release that alters gut response, or adverse reactions to chemical additives present in foods.

Symptoms may occur following the consumption of a food that was previously well tolerated. This is due to the interaction between the microbiota, food-specific antigens, and the immune response, which leads to the release of numerous inflammatory mediators. These mediators also have broader effects, including the activation of nociceptive neurons and alterations in mood. This highlights the rationale behind why diets such as the IgG-based elimination diet may provide benefits for certain patients [68].

Nevertheless, this is a response that simultaneously involves and intersects multiple integrated systems, so it is unrealistic to expect that such a strategy could completely eliminate all symptoms. For example, a possible approach could be to combine the elimination diet with a low-FODMAP diet in cases of persistent symptoms [63]. To date, despite encouraging results from two randomized trials [60,65], evidence supporting the efficacy of IgG-guided elimination diets remains limited. The lack of standardization in IgG testing, the absence of long-term data, and the lack of meta-analyses on this approach have led several scientific societies to discourage its routine clinical use, except within experimental contexts.

### 2.4. Diet Based on Soluble and Insoluble Fiber

The recommendation from the National Academy of Sciences Institute of Medicine for adults is to consume about 20–35 g of fiber per day, although on average intake commonly falls well below this range [69].

Dietary fiber is a non-starch polysaccharide derived from plant-based foods and is poorly digested by human enzymes. It is found in cereals, fruits, vegetables, as well as in seeds, nuts, and legumes [70]. Although there is no single definition of fiber, it is generally agreed that the term refers to carbohydrates that are neither hydrolyzed nor absorbed in the upper gastrointestinal tract. It is, of course, necessary to distinguish this class of non-digestible carbohydrates from digestible and absorbable carbohydrates that are glycemic, such as sugars and starch [69].

Dietary fiber is commonly classified as soluble or insoluble. The former (found in fruits, cereals, etc.) forms viscous solutions that delay gastric emptying and absorption in the small intestine and are fermented by bacteria in the proximal colon to a greater extent than insoluble fiber. Insoluble fibers (cellulose, hemicellulose, lignin), on the other hand, have less effect on the viscosity of intestinal contents. Their physiological effect is to retain water and increase the size and volume of stool, and they are fermented to a very limited extent.

The use of a fiber-rich dietary treatment is associated with numerous benefits in patients with IBS, although differences exist in the use of the two types of fiber [70].

The fermentation of dietary fibers by the gut microbiota leads to the production of short-chain fatty acids (SCFAs) (mainly propionate, butyrate, and acetate), which are utilized by the microorganisms themselves. Generally, these SCFAs improve oxidative stress, inflammation, and apoptosis of intestinal mucosal cells. Additionally, they induce a decrease in visceral sensitivity through the stimulation of intraluminal serotonin [71].

Regarding the dietary treatment of IBS, dietary fibers appear to be associated with improvements across all aspects, such as abdominal pain, bloating, digestion, and bowel habit alterations. They specifically act on the nervous system of the gut–brain axis, cause changes in pH (acidification due to SCFA), affect luminal intestinal pressure, and stimulate the release of serotonin [72].

Moreover, the first recommendation for a patient suffering from chronic constipation is often the adoption of a fiber-rich diet. Therefore, this approach could be particularly useful for patients with IBS-C.

However, the use of dietary fiber for the treatment of IBS remains controversial, especially when implemented through foods. This is because, although some studies show symptom improvements with both low-fiber and high-fiber diets, others report symptom worsening due to the high FODMAP content in these food sources. The symptoms most frequently reported to worsen are flatulence, bloating, and abdominal pain [69].

Other studies emphasize that, in any case, fiber plays a fundamental role in the prevention of this condition.

A case–control study (90 cases and 355 controls) assessed individuals’ dietary intakes using a food frequency questionnaire (FFQ), finding evidence that a high dietary fiber intake was inversely correlated with typical IBS symptoms, and therefore associated with a lower risk of developing the condition [72]. A cross-sectional study including 998 Iranian adolescent girls (16% with IBS), based on a food frequency questionnaire (FFQ), identified a negative association between fiber intake and the likelihood of having IBS. In particular, healthy subjects were those who consumed a significantly higher amount of soluble fiber [73].

Thus, it has often been emphasized that fiber supplementation in patients with IBS leads to an improvement in overall symptoms. While insoluble fiber increases stool volume, soluble fiber, which undergoes enzymatic digestion by the microbiota, promotes an increase in the percentage of short-chain fatty acids in the stool, which may help nourish the intestinal mucosa and improve mucus production.

Therefore, although not yet fully understood, some studies suggest that by altering intestinal transit time, fiber is capable of alleviating IBS symptoms [74].

We can therefore conclude that this is a highly controversial topic, with the debate focusing not only on the use of fiber within a dietary treatment for IBS but also on the use of soluble versus insoluble types. The majority of studies conducted have examined fiber in general, without considering this subdivision. However, soluble and insoluble fibers behave differently at the intestinal level, and the former are considered especially beneficial in the early stages of treatment, during which the intake of insoluble fibers may even worsen symptoms [75].

Further evidence of the positive effect of soluble fiber comes from dietary interventions with FOS and GOS, which have been shown to enrich the gut microbiota, particularly *Bifidobacterium* and *Lactobacillus* spp. [76].

Therefore, it is primarily products containing soluble fiber (especially FOS and GOS taken daily) that provide benefits, whereas no significant evidence has been found to support the usefulness of products containing insoluble fiber, such as wheat bran [77].

To date, the most widely accepted dietary recommendation for IBS is the low-FODMAP diet, as we have already seen. However, this may lead to inadequate fiber intake in patients, so supplementation could be helpful, especially in the very early stages [78].

New frontiers are also being explored, such as enriching common Western diet foods like pasta with fibers beneficial for IBS, which could become part of the diet for individuals following a low-FODMAP diet. Recent research has identified fibers such as cellulose and guar gum that are not classified as FODMAP and can be consumed by IBS patients without triggering typical symptoms. These fibers are characterized by low fermentability, insolubility, and increased viscosity [79].

Once again, it emerges that a personalized approach, tailoring dietary fibers with different characteristics according to the various IBS subtypes, could make this strategy even more beneficial and represent a breakthrough for the patient’s quality of life.

### 2.5. The Lactose-Free Diet

Lactose intolerance and IBS often present with similar gastrointestinal symptoms, leading many people to wonder about a possible correlation. Those suffering from these disorders often believe that lactose is one of the primary factors related to the pathophysiology and tend to eliminate it from their diet.

Although subjective lactose intolerance is much more common in patients with IBS than in the general population, objective lactose malabsorption, measured through specific tests, is not significantly higher in IBS patients compared to healthy controls. These patients often report symptoms following the consumption of dairy products, but only a subgroup of them truly suffers from lactose malabsorption, suggesting that other factors beyond lactase deficiency contribute to the symptoms in question [80,81,82,83].

Some studies have highlighted increased immune activation, visceral hypersensitivity, and anxiety in IBS patients who develop symptoms following lactose ingestion, even without diagnosed malabsorption. These factors could be responsible for an amplification of symptoms and the individual perception of intolerance [82,84]. Moreover, it appears that unabsorbed lactose fermented by the microbiota produces metabolites that may contribute to the symptoms, potentially linking gut microbiota activity to both lactose intolerance and the symptoms of IBS [85].

Based on a subjective assessment of the symptoms experienced, patients often tend to eliminate lactose from their diet. Symptoms of lactose intolerance and IBS are often difficult to distinguish. Furthermore, it should be considered that patients with lactase deficiency have symptoms that depend on the lactose dose and lactase expression, as well as on the microbiota and sensitivity of the gastrointestinal tract [86].

The lactose-free diet is often recommended to patients with IBS as well, given its easy adherence and the wide availability of lactose-free products on shelves nowadays [80].

It involves minimizing or even completely eliminating lactose from the diet, replacing it with lactose-free or plant-based alternatives [87].

It is important that individuals following this diet are properly educated on the importance of reading labels (as some foods may contain traces of lactose) and on the need to obtain calcium, vitamin D, and other essential micronutrients found in dairy products through lactose-free fortified foods or supplements, to prevent malnutrition. Additionally, some individuals may be able to tolerate small amounts of lactose. Individual tolerance can be tested through a gradual reintroduction of small quantities and monitoring of symptoms [87,88].

Some studies suggest that a lactose-free diet can be effective in reducing IBS symptoms. This is because lactose is one of the fermentable carbohydrates and is therefore definitely one of the triggers of symptoms, especially in cases of lactose intolerance [89,90].

However, although this dietary approach may improve symptoms in patients with IBS and concurrent lactose malabsorption, there is insufficient evidence to support its routine use in all IBS patients. If a patient reports sensitivity to milk without an objective diagnosis of lactose malabsorption, it would be more appropriate to follow a milk-free diet rather than a lactose-free diet [80].

Restricting lactose alone may lead to marginal benefits, whereas considering it as part of a low-FODMAP diet could be more effective in addressing a broader range of symptoms. The guidelines from the British Dietetic Association recommend a balanced diet and a healthy lifestyle as the first-line treatment, with the application of the low-FODMAP diet as a second-line treatment [91].

In conclusion, to manage a proper individualized treatment, it is essential to differentiate between IBS and lactose intolerance. Patients diagnosed with IBS may often have undiagnosed lactose intolerance, and vice versa [92]. To achieve this, objective tests are obviously needed, such as the hydrogen breath test, which allows for distinction between the two conditions and the development of the most appropriate dietary recommendations [80,92].

### 2.6. The Gluten-Free Diet

Celiac disease (CD) is a permanent T cell-mediated enteropathy caused by the ingestion of gluten, a collective term for the proteins found in wheat, rye, and barley. These proteins contain a high proportion of prolamins, which drive the adaptive immune response in CD [93]. The main causes of CD can be divided into genetic and environmental factors; risk factors include the presence of HLA-DQ2/DQ8 genes, viral infections, alterations in the gut microbiota, and a family history of celiac disease [94].

The gluten-free diet (GFD) involves eliminating all gluten-containing grains, such as wheat, rye, barley, oats, spelt, kamut, or their hybrid strains. Naturally gluten-free grains like rice, corn, quinoa, millet, and amaranth are allowed. Oats, if certified gluten-free, can be consumed, although some varieties may still trigger an immune response in individuals with celiac disease [95]. The diet is generally safe, but it may lead to long-term nutritional deficiencies, particularly in fiber, iron, calcium, folate, and vitamins. Recent research highlights that gluten-free (GF) products still have nutritional shortcomings, with lower protein content and higher levels of fat and salt compared to their gluten-containing counterparts. However, improvements have been observed in fiber and sugar content. Efforts from experts and the food industry are needed to enhance the nutritional quality of GF products by using alternative ingredients and technologies to enrich them with micronutrients and fiber [96].

In recent years, the consumption of gluten-free products has increased significantly, even among individuals without a diagnosed health condition. This trend is often driven by the desire for a healthier lifestyle, although such benefits are more likely associated with the reduction in processed foods rather than the elimination of gluten itself. The widespread adoption of the GFD is often fueled by advertising, frequently lacking real scientific support [97].

Gluten metabolism and the gut microbiota are interrelated, and their interactions help define intestinal health and its homeostasis. The presence or absence of gluten in the diet can alter the diversity and proportions of microbial communities within the gut [98].

In addition to patients with CD, the GFD is also used by patients with IBS, as individuals often attribute their symptoms to components found in gluten-containing foods as well [99]. The cluster of symptoms associated with IBS, such as diarrhea, constipation, or abdominal pain, can overlap with those of gluten-related disorders [100] and it is recommended that patients presenting with IBS-like symptoms, particularly those with diarrhea, undergo screening for CD [101].

Two randomized clinical trials have examined the effect of the GFD in patients with IBS. The first study (34 patients) found that IBS patients who had good symptom control while following a GFD experienced a significant worsening of symptoms when gluten was reintroduced. This suggests that gluten may be a trigger for some IBS patients. The second study (72 patients, double-blind, placebo-controlled) involved patients following a GFD for 6 weeks, after which they were divided into two groups: one received powdered gluten and the other a placebo. In the gluten group, all symptoms initially improved with the GFD but then significantly worsened after just one week of gluten reintroduction [102]. The GFD has been studied as a potential treatment for IBS-D. Some studies have shown that IBS-D patients positive for specific biomarkers experienced improvements in stool frequency and gastrointestinal symptoms after 6 months on a GFD. Another study found that 71% of IBS-D patients reported improvement after 6 weeks of following a GFD [103]. Finally, a study examined the effectiveness of three different diets in improving the quality of life of patients with IBS: a low-FODMAP diet, a GFD, and a balanced diet. Each diet was followed for 4 weeks by a group of patients. The results showed that all three diets led to a significant reduction in symptom severity, bloating, and abdominal pain, as well as an improvement in quality of life. However, only 11% of patients preferred the GFD [41].

Moreover, when comparing the side effects of gluten-free products with those of approved medications for the treatment of IBS, along with their high costs, it becomes clear that a treatment based on a GFD is significantly less toxic. Patients can easily adapt their personal dietary preferences by following simple guidelines, making the GFD an accessible and sustainable option for symptom management [104].

Currently, there is still insufficient evidence to recommend a GFD for reducing IBS symptoms, as symptom improvement may be due not only to gluten exclusion but also to the limited intake of other foods containing, for example, fermentable sugars [105].

In any case, diet plays a fundamental role in the management of IBS, and it is important that patients are aware of what they eat and how it affects their symptoms. There are still many areas of research to explore, particularly regarding the various dietary approaches. Physicians and nutritionists must consider multiple factors to provide more personalized and safe recommendations, and it is important to avoid interventions that could lead to nutritional deficiencies [106].

### 2.7. Influence of Diet on the Gut Microbiome, Metabolome, Neurohormonal Pathways, Clinical Outcomes, and Psychological Health

Based on current knowledge, we can clarify that the nutritional approach in functional disorders represents a key tool in the multidimensional management of these patients, capable of influencing the composition of the microbiota [107], intestinal metabolic activity, the production of neuro-hormonal mediators, symptom perception, and indirectly, psychological aspects as well [108].

### 2.8. Diet and Gut Microbiota

The gut microbiome may play a central role in maintaining gastrointestinal health due to its intricate and diverse microbial composition, but it is also thought to play a key role in the pathogenesis of DGBI, such as IBS, functional abdominal bloating, and functional abdominal distension [109,110].

Specifically, the ingestion of fermentable foods with low intestinal absorption, such as FODMAPs, can promote fermentation by gas-producing bacteria, contributing to abdominal distension, flatulence, and visceral hypersensitivity [111]. It has been observed that patients with IBS often exhibit an altered microbial composition, with an increase in genera such as *Dorea*, *Ruminococcus* [112], and *Clostridium*, and a reduction in beneficial populations like *Bacteroidetes*, *Bifidobacterium*, and *Faecalibacterium prausnitzii* [107]. This dysbiosis can facilitate low-grade inflammatory processes and increase intestinal permeability, worsening the symptoms [107,113,114].

In light of these observations, interest has increased in therapeutic strategies aimed at modulating the microbiota, including the use of probiotics, prebiotics, and, in selected cases, fecal microbiota transplantation [115].

Furthermore, it has been hypothesized that small intestinal bacterial overgrowth (SIBO) may represent a relevant pathogenic mechanism in a subpopulation of patients with IBS, contributing to excessive gas production and symptoms of bloating and postprandial pain [116,117].

### 2.9. Effects on the Metabolome

Despite the clinical benefits achieved in managing IBS symptoms, the low-FODMAP diet has been shown to significantly alter the composition of the gut microbiota and the metabolic profile [118]. However, it is still unclear how long these alterations persist and what their long-term impact on overall health may be [119]. Unlike gluten, FODMAPs appear to be associated with specific changes in the microbiota, linked to the production of metabolites derived from phenolic compounds and 3-indolepropionate [120], substances considered beneficial for the body. This highlights the need to balance the positive effects on symptom management with the potential consequences for gut microbiota health.

Moreover, alterations in the microbiota and metabolome are frequently observed in patients with functional gastrointestinal disorders, supporting the hypothesis that microbial composition may influence the response to currently available therapies. In this context, ingested nutrients are transformed by the microbiota into a wide range of bioactive metabolites, collectively referred to as the gut metabolome. Among these, SCFAs such as acetate, propionate, and butyrate [121] play a crucial role due to their anti-inflammatory effects, trophic functions for the intestinal mucosa, and immunomodulatory properties [122,123].

Conversely, unbalanced dietary patterns, such as those excessively rich in animal proteins or refined sugars, can promote the production of potentially toxic metabolites, including ammonia, sulfides, and indole. These compounds may compromise the integrity of the intestinal barrier, negatively influence the local immune response, and alter the psychological perception of the disorder itself [124]. A key role is also played by tryptophan metabolism, the precursor of serotonin, whose regulation depends on both diet and the composition of the gut microbiota [125]. Alterations in this metabolic pathway can affect mood tone, intestinal motility, and the perception of visceral pain, making the metabolome a central component of the gut–brain axis [120,126,127].

### 2.10. Neuro-Hormonal Signals and Symptom Perception

The presence of nutrients in the intestinal lumen activates a complex network of neuroendocrine signals, mediated by enteroendocrine cells, enteric neurons, and visceral afferent fibers.

For example, recent evidence has identified neuropod cells, a synaptically active subset of enteroendocrine cells, as key mediators of gut–brain communication. These cells convert nutrient signals into neural activity via glutamatergic synapses with afferent neurons [128]. Altered signaling involving guanylyl cyclase C (GUCY2C) and neuropod cells has been linked to visceral hypersensitivity in IBS models [129,130], suggesting that specific nutrients may influence symptoms through direct effects on gut sensory pathways.

Hormones such as cholecystokinin (CCK), glucagon-like peptide-1 (GLP-1), and peptide YY (PYY), released in response to nutrients, regulate satiety, motility, and intestinal secretion [131]. These cells, distributed throughout the gastrointestinal tract, respond selectively to macronutrients such as fats, carbohydrates, and proteins, and activate local or vagal neural circuits that contribute to appetite control and digestion [132,133,134].

The stimulation of mucosal mechanoreceptors and chemoreceptors by specific nutrients—particularly in the presence of dysbiosis—can amplify nociceptive signaling, contributing to sensations of pain and bloating. Intestinal dysbiosis, in fact, alters the production of microbial metabolites such as SCFA, which in turn affect hormonal secretion and visceral sensitivity [131,135,136]. In predisposed individuals, these alterations may promote a hypersensitive response to normal intestinal stimuli, facilitating the development of chronic abdominal pain and distension [137].

Visceral hypersensitivity represents one of the main pathophysiological mechanisms underlying IBS and is exacerbated by imbalances in neuro-hormonal signals triggered by the presence of nutrients. This condition can be further influenced by psychological factors and stress, which modulate gut reactivity through the hypothalamic-pituitary-adrenal (HPA) axis, contributing to heightened symptom perception [138,139], as well as to obesity and metabolic disorders [140]. However, these mechanisms can be modulated through personalized dietary interventions. Such approaches can reduce intestinal fermentation, improve mucosal barrier function, and regulate neuroendocrine activity [135,136].

### 2.11. Clinical Response and Symptomatology

Several dietary regimens have demonstrated effectiveness in improving IBS symptoms. Among these, the low-FODMAP diet currently has the strongest scientific evidence: it significantly reduces pain, bloating, and bowel irregularities in approximately 70% of patients, although its impact on microbiota diversity warrants caution regarding long-term use [141,142]. In contrast, the Mediterranean diet, which is more sustainable over the long term, exerts a protective effect due to its richness in fiber, polyphenols, and unsaturated fats. It favorably modulates the gut microbiota and contributes to the reduction in inflammation and improvement in quality of life [143,144].

However, the clinical efficacy of dietary interventions is closely linked to adherence and personalization: genetic, psychological, metabolic, and microbiological factors necessitate an individualized approach to maximize benefits.

### 2.12. Psychological Impact and the Gut–Brain Axis: Focus on the Relationship Between IBS, the Low-FODMAP Diet, and Eating Disorders

The influence of diet in IBS extends beyond the physiological dimension, involving the gut–brain axis and the psychological sphere [145]. The bidirectional connection between the gut and the brain implies that intestinal dysbiosis may contribute to the onset or worsening of psychiatric disorders such as anxiety and depression [146], frequently observed in patients with DGBI [147,148]. In this context, diet represents a powerful modulator of the microbiota, systemic metabolism, and neuro-endocrine response, and thus a fundamental therapeutic leverage in managing these disorders.

A recent observational study (Sultan et al., 2024) [149] specifically investigated the presence of disordered eating behaviors in patients with IBS. In a sample of 202 IBS patients and 109 controls, 33% of the IBS subjects scored ≥ 2 on the Sick, Control, One stone, Fat, Food (SCOFF) questionnaire, indicative of eating disorders, compared to 16% in the control group (*p* < 0.001). Additionally, mean scores on the Eating Habits Questionnaire (EHQ), which assesses orthorexic tendencies, were significantly higher in IBS patients (82.9 ± 18.1) than in controls (73.5 ± 16.9; *p* < 0.001). These findings suggest an increased vulnerability to eating disorders, including orthorexia, among individuals with IBS.

However, unsupervised restrictive diets, such as the low-FODMAP diet, although effective in reducing symptoms in a significant portion of patients, carry the risk of generating or worsening disordered eating behaviors [150]. Among these, orthorexia (a pathological obsession with “healthy” eating that leads to avoiding entire food groups) and avoidant/restrictive food intake disorder (ARFID) are particularly relevant [151,152]. Patients with ARFID avoid certain foods or food groups to the extent that they develop malnutrition, weight loss, and in more severe cases, require artificial nutritional support, such as enteral or parenteral feeding [153,154].

This risk is particularly high in individuals with IBS or other DGBI, who may exhibit greater psychological vulnerability. The coexistence of eating and nutritional disorders (ENDs) can compromise the effectiveness of dietary therapy and worsen overall clinical outcomes [152]. Although the relationship between eating disorders and IBS is well documented, it is still often underestimated: numerous studies show that functional gastrointestinal symptoms are common in patients with eating disorders, such as anorexia nervosa, and these symptoms can in turn contribute to the persistence of the eating disorder, creating a vicious cycle [148,155,156].

For example, patients with anorexia nervosa often experience chronic constipation, the severity of which is related to the duration of the disorder and the degree of malnutrition [157]. This condition, which can result from insufficient food intake and electrolyte imbalances, tends to improve only after at least three weeks of adequate nutritional rehabilitation [158]. Therefore, it is essential that any dietary regimens are prescribed only after a thorough clinical and nutritional assessment, taking into account the risk of eating disorders [157].

In light of this complexity, a multidisciplinary therapeutic approach is essential, involving gastroenterologists, clinical dietitians, general practitioners, and mental health specialists. Collaboration with an experienced clinical nutritionist, preferably a Registered Dietitian Nutritionist (RDN), ensures integrated care that addresses both the organic and metabolic aspects as well as the psychological and behavioral components of the patient. Only through the synergy of personalized interventions and multidisciplinary specialist support is it possible to achieve lasting symptomatic benefits and improve the quality of life for patients affected by IBS.

### 2.13. The Role of Beverages in IBS Management

Beverages can significantly influence the management of IBS by modulating intestinal motility, visceral sensitivity, and microbiota composition.

Adequate hydration is essential in managing IBS subtypes; it can help prevent dehydration in patients with IBS-D, while for those with IBS-C, proper hydration is crucial to facilitate intestinal transit and prevent constipation [159,160]. Additionally, it has been observed that the intake of cold water may lower the visceral pain threshold in IBS patients, intensifying abdominal symptoms compared to warm water, which appears to be better tolerated [161].

The consumption of coffee shows conflicting effects: while some studies suggest that intake may reduce the risk of developing IBS [162], others highlight a positive association between the consumption of coffee, as well as alcohol and artificial sweeteners, and the development of IBS or the onset of symptoms in IBS patients [163,164].

Carbonated beverages are often associated with IBS-D [165]. Fermented drinks such as kombucha, enriched with inulin and vitamins, have shown benefits in women with IBS-C by improving bowel movement frequency and stool consistency; however, further validation of these claims through additional clinical studies is needed [166].

As previously mentioned, lactose-containing beverages can trigger gastrointestinal symptoms in patients with lactose intolerance, a condition common in individuals with IBS, although the two should not be confused. Sugary drinks high in FODMAP, particularly those rich in fructose or sorbitol, can induce symptoms such as bloating, flatulence, and diarrhea due to their osmotic fermentation in the colon [167,168].

Therefore, a thoughtful selection of beverages and daily hydration, based on patient phenotyping and individual tolerance, is a key element in the dietary management of IBS and should be considered an integral part of a multidisciplinary nutritional approach.

### 2.14. Macro and Micronutrients

According to the ESPEN guidelines on hospital nutrition, a balanced diet should provide 50–60% of total energy from carbohydrates, 30–35% from lipids, and 15–20% from proteins. Carbohydrates are the main source of energy for the body, supporting brain and muscle functions. Proteins are essential for the synthesis and maintenance of muscle mass, as well as playing a crucial role in the immune response. Fats, in addition to providing energy, are fundamental for the absorption of fat-soluble vitamins and hormone production. An adequate balance of these macronutrients is vital to maintain metabolic health and prevent malnutrition, especially in clinical settings [169].

Trace elements and vitamins, collectively called micronutrients (MNs), are essential for human metabolism. The importance of MNs in common diseases is recognized by recent research, with deficiencies significantly affecting prognosis [170].

Studies have shown inadequate intake of vitamins A, B2, B9, B12, calcium, and zinc in the habitual diet of patients with IBS [171]. Vitamin D deficiencies are frequently observed in IBS patients [172,173]. One study indicates that there is no evidence to support the use of vitamin D for managing IBS symptoms; however, the high prevalence of vitamin D deficiency suggests that routine screening and supplementation should be implemented in this population for general health reasons [173]. Nevertheless, other studies report conflicting results, showing benefits of vitamin D supplementation on IBS symptoms [174,175].

Magnesium is an essential mineral for optimal metabolic function [176], and the diet of patients with IBS has been found to be deficient in it [177]. Magnesium oxide has been clinically used as a laxative for many years. The increasing availability of new drugs for treating constipation has led to emerging scientific evidence supporting the use of magnesium oxide, which is convenient to administer, low-cost, and safe. Although recent randomized controlled trials (RCTs) have demonstrated that magnesium oxide is safe and effective for treating constipation, evidence of its efficacy for treating symptoms of irritable bowel syndrome, particularly in the constipation-predominant subgroup, still needs to be established [178].

In patients with IBS, iron deficiency may also occur, with lower levels compared to individuals without irritable bowel syndrome [171]. Dietary iron absorption is crucial for health and essential to prevent iron deficiency states and related comorbidities such as anemia [179]. However, oral iron supplementation can cause constipation, nausea, and diarrhea [180].

Zinc is an essential trace element for living organisms, and zinc homeostasis is crucial for maintaining normal physiological functions of cells and organisms. Both excess and deficiency of zinc, as well as imbalances in zinc homeostasis, are associated with various intestinal diseases such as inflammatory bowel disease, colorectal cancer, and IBS. In IBS, zinc may regulate the central nervous system and the gastrointestinal system by influencing intestinal mucosal barrier function, mast cell activation, and neurotransmitter secretion, all of which are related to the onset and development of IBS [181].

Similarly, patients with IBS have been found to have significantly lower serum selenium levels compared to healthy controls, and selenium deficiency may trigger IBS symptoms; therefore, selenium supplementation should be considered for patients with IBS [182].

### 2.15. Dietary Supplements in IBS: Probiotics

An individualized dietary guide combined with probiotics has shown great potential to regulate intestinal motility and improve symptoms [183].

Probiotics are live microorganisms that, when administered in adequate amounts, confer a health benefit to the host. The overall benefit of probiotics is to support a healthy digestive tract and immune system by creating a more favorable intestinal environment [184]. The gut microbiota consists of 17 families, 50 genera, and more than 1000 bacterial species; its composition varies among individuals and changes throughout life. As already outlined, most data support the idea that the microbiota composition is different in patients with IBS compared to healthy subjects [185]. Studies show that patients with IBS have a low population of *Lactobacillus* spp. and *Bifidobacterium* spp. [186]. Managing the gut microbiota is a hot topic in IBS treatment because some studies have shown promising results, although there is a lack of solid data to provide clear evidence regarding which bacterial species to use, whether individually or combined, and the duration of treatment [187].

A review suggests that taking probiotics at either low or high doses for less than 8 weeks significantly improves IBS symptoms and quality of life. Single-strain probiotics have been found to be more effective in reducing symptoms, with *Bifidobacterium infantis*, *Saccharomyces cerevisiae*, and *Lactobacillus plantarum* appearing to provide the greatest benefits to patients [188]. In addition, a recent meta-analysis by Ford et al. [189], including 53 randomized controlled trials, confirmed that probiotics significantly improve global IBS symptoms compared to placebo, although results varied depending on the strain and formulation used. This provides further evidence supporting the use of probiotics as part of a personalized therapeutic strategy in IBS patients [189].

Overall, probiotics are effective, safe, well-tolerated, and well-recommended for treating different types of IBS, but only for a limited time, although further confirmation from more studies is still needed [190].

Finally, additional research is necessary to confirm the role of probiotics and dietary fibers across various clinical subgroups and to fully characterize their effects on the gastrointestinal microbiota [191].

### 2.16. NICE Recommendations and the Importance of Specialist Involvement in the Management of Patients with IBS

The National Institute for Health and Care Excellence (NICE) provides dietary advice and lifestyle modification guidelines for the initial management of IBS. Specifically, patients are advised to consume regular meals, eat slowly and mindfully, and drink at least eight glasses of fluids daily, preferably water or caffeine-free beverages. Caffeine intake should be limited to a maximum of three cups per day. Additionally, reducing the consumption of alcohol and carbonated drinks is recommended, as these may exacerbate symptoms such as bloating and diarrhea. Regular physical activity is also encouraged to improve overall well-being. If symptoms persist despite these changes, referral to an experienced dietitian is indicated. At this stage, a low-FODMAP diet may be considered, always under specialist supervision, both due to its restrictive nature and the potential risk of nutritional deficiencies [192], and because structured FODMAP education provided by dietitians appears to enhance patients’ self-efficacy and improve long-term symptoms without compromising overall FODMAP intake [193]. The initial dietary and lifestyle recommendations from NICE represent a fundamental first step in IBS management; however, given the complexity of the syndrome, an integrated multidisciplinary approach is often required.

Functional gastrointestinal disorders are common and costly for the healthcare system. Most specialized care is provided by gastroenterologists, but only a minority of patients experience symptom improvement.

Integrated multidisciplinary clinical care (including gastroenterologists, dietitians, nutrition biologists, psychiatrists, behavioral therapists such as pelvic floor physiotherapists, and gut-focused hypnotherapists) appears to be superior to care provided solely by a gastroenterologist in relation to symptoms, specific functional disorders, psychological status, quality of life, and healthcare costs for the treatment of functional gastrointestinal disorders. Offering multidisciplinary care should be considered for patients with functional gastrointestinal disorders [194].

It is clear that not all patients with IBS are the same. The multidimensional nature of this condition, which presents with diverse pathophysiology and clinical features, highlights the need for interdisciplinary solutions. There is no one-size-fits-all approach when it comes to treatment plans for patients with IBS [195].

**Table 3 nutrients-17-02496-t003:** Comparison of various diets highlighting their pros and cons.

	Low-FODMAP Diet	Mediterranean Diet	Gluten-Free Diet	Lactose-Free Diet	High-Fiber Diet	IgG-Guided Diet
Goal	Reduce fermentation and FODMAP-related symptoms	Balanced, anti-inflammatory nutrition	Reduce symptoms similar to celiac disease	Reduce symptoms related to lactose intolerance	Increase stool bulk, regulate motility	Eliminate foods with IgG-mediated immune response
Physiopathological Rationale	Eliminates fermentable carbohydrates	Rich in protective nutrients and natural prebiotics	Some IBS patients show sensitivity to gluten	Lactose intolerance is common in IBS	Fibers improve intestinal function	Based on food-specific IgG immune response
Effects on IBS	Significant symptom improvement	Benefits on microbiota and inflammation	Reduced symptoms in some patients (mainly IBS-D)	Reduced bloating and diarrhea in intolerant individuals	Particularly useful in IBS-C	Some studies show benefits, but evidence is inconclusive
Limitations /Criticisms	Restrictive and difficult to maintain	Naturally contains FODMAP (e.g., legumes, fruits)	Effective only in subgroups; possible placebo effect	Only effective in patients with actual intolerance	May worsen symptoms in IBS-D	Limited data; risk of excessive dietary restrictions
Composition	Temporary exclusion, followed by FODMAP reintroduction	Whole grains, vegetables, fruits, legumes, EVO oil	Excludes all gluten-containing products	Eliminates milk and lactose-containing dairy products	Rich in soluble and insoluble fibers	Personalized exclusion based on IgG tests
Tolerability	Good in the short term; challenging long-term	Generally good, variable in IBS	Good in those reporting gluten sensitivity	Good in lactose-intolerant individuals	Good in IBS-C; poor in IBS-D	Variable; depends on individual response
Scientific Support	Solid and supported by international guidelines	Strong in general nutrition; emerging in IBS	Limited; useful in selected non-celiac patients	Limited but recommended when positive for intolerance	Recommended for IBS-C	Controversial; studies have methodological limitations
Effects on Microbiota	Risk of reduced diversity if not balanced	Positive (prebiotic effect)	Potential reduction in diversity if not well managed	Less impact, but imbalance possible if poorly compensated	Promotes beneficial bacteria if tolerated	Unclear or poorly studied

**Note:** The indicators of tolerability and scientific support are derived from a narrative synthesis of the current literature reviewed in this article, including randomized controlled trials, meta-analyses, and international guidelines. No validated scoring system currently exists for the comparative assessment of dietary strategies in IBS. These indicators should therefore be interpreted as qualitative summaries rather than quantitative ratings.

## 3. Discussion

In the current landscape of IBS management, dietary intervention represents one of the most promising yet complex therapeutic tools to standardize. The growing body of scientific evidence has clearly shown that no single diet is universally effective for all patients, given the considerable clinical and pathophysiological heterogeneity of this condition. In light of such complexity, this review offers not only a critical analysis of the main nutritional strategies used in IBS but also a methodological and clinical reflection on how diet is prescribed, followed, and perceived by patients.

Among all the approaches examined, the low-FODMAP diet remains the one with the strongest scientific support. It is based on reducing the intake of short-chain fermentable carbohydrates, which are responsible for hyperosmolarity, bacterial fermentation, and intestinal lumen distension. Data show a significant reduction in bloating and abdominal pain in about 70% of patients, regardless of IBS subtype. However, the limitations of this approach become evident during the maintenance phase, as prolonged restriction may reduce microbiota biodiversity—particularly of beneficial species such as *Bifidobacterium*—thereby increasing the risk of dysbiosis. The low-FODMAP diet requires competent professional support during the reintroduction and personalization phase, otherwise, it may result in therapeutic failure or the development of dysfunctional eating behaviors.

An opposing model in terms of philosophy and applicability is the Mediterranean diet. Rich in fiber, polyphenols, and monounsaturated fats, and low in ultra-processed foods, it offers systemic anti-inflammatory effects and a positive impact on gut microbiota, promoting the abundance of beneficial strains and the production of SCFA. However, despite its effectiveness in improving overall health and quality of life, the Mediterranean diet includes many high-FODMAP foods (such as a wide variety of fruits, vegetables, and legumes), which may exacerbate symptoms in some patients. Therefore, this approach also requires careful personalization, modulating both quantity and timing of intake, without abandoning the core principles of the Mediterranean lifestyle.

The gluten-free diet, originally designed for patients with celiac disease, has also been applied to individuals with IBS, especially in cases of suspected non-celiac gluten sensitivity. However, a significant portion of the benefit observed may actually result from the reduction in fructans found in gluten-containing cereals, rather than from the elimination of gluten itself. Some controlled studies suggest a placebo or nocebo effect associated with gluten reintroduction. This diet may indeed be helpful for specific subgroups of patients, but it is essential to clearly distinguish between celiac disease, non-celiac sensitivity, and IBS to avoid unnecessary restrictions.

A similar consideration applies to the lactose-free diet, often empirically adopted by IBS patients without a confirmed diagnosis of lactose malabsorption. Although lactose intolerance is more frequently reported in IBS patients, objective malabsorption, confirmed via clinical testing, is not more prevalent than in the general population. Therefore, patients should undergo a hydrogen breath test before removing dairy products from their diet to avoid deficiencies in calcium, vitamin D, and other micronutrients. In patients with confirmed intolerance, the lactose-free diet represents a simple, effective, and widely accessible option.

The role of a high-fiber diet is different and has historically been recommended for managing IBS, especially in the constipation-predominant subtype. However, not all fibers are the same. Soluble fibers, such as psyllium, have proven effective in improving stool consistency, reducing bloating, and enhancing intestinal transit. On the other hand, insoluble fibers may worsen symptoms, especially in IBS-D patients, by increasing bloating and urgency. Moreover, many fiber-rich foods are also high in FODMAP, creating a therapeutic paradox that only a personalized nutritional analysis can resolve.

Lastly, the IgG-based elimination diet represents a controversial yet intriguing approach. Some studies suggest a correlation between elevated serum IgG levels against specific foods and the presence of gastrointestinal symptoms. However, IgG presence may simply indicate immune exposure rather than clinically relevant intolerance. Current evidence is still insufficient for routine recommendation, but in selected cases, especially when the low-FODMAP diet fails, this approach may offer an additional option, provided it is always supervised by a professional.

Beyond dietary evidence, this review also explores integrated pathophysiological aspects, such as the role of the gut microbiota, metabolome, and gut–brain axis. Interactions between nutrients, the microbiome, gastrointestinal hormones, and the central nervous system are now recognized as key in modulating IBS symptoms. Nutrition, therefore, is not merely fuel but also a signal, capable of influencing immune, endocrine, and neurocognitive activity. Moreover, diet affects the production of microbial metabolites (such as butyrate and propionate) that can have anti-inflammatory or trophic effects, or, conversely, harmful consequences when derived from non-physiological substrates or produced in excess.

One often overlooked dimension is the psychological and behavioral one. IBS patients are more vulnerable to developing disordered eating behaviors, particularly in the presence of self-imposed dietary restrictions. For example, the symptomatic effectiveness of the low-FODMAP diet may turn into a pathological obsession with food control, leading to conditions such as orthorexia nervosa or ARFID. These disorders can worsen clinical outcomes, reduce quality of life, and undermine the effectiveness of any therapeutic approach. It is therefore essential to monitor the patient’s relationship with food and intervene early if warning signs emerge.

In response to this complexity and the clinical challenges frequently encountered in practice, this review proposes an original and pragmatic approach: the “10 Dietary Commandments for IBS” (Figure 1). Far from being mere prescriptive rules, these principles represent a critical synthesis of scientific evidence and clinical experience, aimed at preventing the most common errors and promoting truly patient-centered management. The commandments emphasize that no diet should be undertaken without specialist support, that flexibility is a clinical virtue, that dietary treatment should not compromise social life or become an aesthetic obsession, and that nutritional interventions must be part of a broader, multimodal therapeutic plan, not viewed as standalone solutions.

In a cultural context still searching for a “perfect diet for all,” these commandments propose a paradigm shift: from uniformity to individualization, from elimination to balance, from rigidity to sustainability. They do not abandon scientific rigor but rather complement it with a vision that acknowledges the bio-psycho-social complexity of IBS and the uniqueness of each patient.

Ultimately, the challenge is not to find “the right diet” in an absolute sense, but rather to build the most suitable, realistic, and human dietary path for each individual, guided by science, yes, but also by clinical common sense, attentive listening, and respect for the patient as a person.

**A patient with IBS should not approach a diet independently, but be guided by a specialist, at least in the initial stages**. Without expert guidance, there is a risk of nutritional mistakes or harmful approaches.**It is important not to forget the intake of essential micro- and macronutrients**. Nutrient deficiencies can compromise overall health and negate the benefits of the diet.**There is no one-size-fits-all diet, but rather tailor-made diets that must be adapted to each individual patient**. Every individual has different needs, so the diet must be personalized.**A diet should not be static, but flexible and adaptable over time**. Nutritional needs evolve, and the diet must adjust accordingly.**A patient with IBS should not expect an immediate ON/OFF effect from a diet; it is not a temporary or symptomatic treatment, but a long-term approach**. Patience is essential for achieving lasting results.**A diet should not interfere with the patient’s social life; occasional deviations are acceptable if they serve a social or psychological benefit**. An overly rigid diet can lead to isolation and frustration.**A diet should not place an economic burden on the patient**. It must be financially sustainable over time.**A patient with IBS should avoid attributing obsessive empathy and emotions to food**. The relationship with food should be healthy and not driven by negative or compulsive emotions.**A diet should not be subordinated to physical appearance**. The primary goal should be well-being, not just aesthetics.**A diet should not be viewed in isolation as a therapeutic solution, but rather be part of a multimodal therapeutic approach**. To be effective, it must be integrated into a broader plan that considers all therapeutic options (e.g., probiotics, neuromodulators).

## 4. Conclusions

Nutritional management of IBS is a key component of therapy but requires a highly individualized and multidisciplinary approach due to the marked interindividual variability and complex pathophysiology of the disorder.

The main dietary strategies for IBS include the low-FODMAP diet, effective in reducing gastrointestinal symptoms but difficult to maintain long-term and potentially detrimental to microbiota diversity; the Mediterranean diet, balanced and sustainable but containing FODMAP-rich foods that require adaptation; the gluten-free diet, useful only in selected subgroups and often confused with fructan reduction; the lactose-free diet, effective for intolerant individuals but often adopted without proper diagnosis, risking nutritional deficiencies; the high-fiber diet, beneficial especially in IBS-C but potentially worsening symptoms in diarrheal subtypes; and the IgG-based elimination diet, still controversial and lacking robust evidence, to be considered only in targeted cases under strict supervision.

The most innovative contribution of this review is the proposal of the “10 Dietary Commandments for IBS”, practical principles designed not to prescribe what to eat, but to guide both patients and clinicians in how to manage dietary interventions. These commandments discourage rigidity, self-prescription, and social isolation, while promoting flexibility, personalization, and long-term sustainability.

Ultimately, the goal is not to identify a universal “perfect diet,” but to build a tailored, realistic, and integrative nutritional pathway, where the 10 commandments serve as a concrete and accessible guide to enhance adherence, improve well-being, and support quality of life for patients living with IBS.

In the context of personalization, one of the main challenges is the lack of reliable and easily applicable clinical predictors to anticipate individual responses to dietary interventions [150]. Future research should therefore focus on the development and validation of simple clinical tools capable of integrating gastrointestinal symptoms, microbiological profiles, and psychological factors in order to guide therapeutic decisions more precisely and sustainably. Furthermore, the adoption of eHealth tools dedicated to nutritional education and symptom monitoring represents a promising complementary resource in the clinical management of IBS, especially for patients with complex needs [196].

While this narrative review offers a critical and integrated overview of the main dietary strategies for managing IBS, it also carries inherent limitations related to its design, which may introduce selection and interpretation bias. The absence of a quantitative meta-analysis prevents a comparative weighting of the evidence, and despite rigorous selection criteria, the inclusion of studies may have favored more recent publications or those aligned with the review’s patient-centered perspective. Furthermore, several of the indications discussed (particularly those related to psychological–behavioral aspects and the impact of dietary patterns on the gut microbiota) are based on preliminary studies or those involving small sample sizes, limiting the generalizability of findings. The “10 Dietary Commandments for IBS,” although innovative and pragmatically derived from clinical experience, have yet to undergo systematic validation through prospective trials or implementation studies. Future research should aim to evaluate their real-world effectiveness in improving adherence, clinical outcomes, and quality of life. These commandments should thus be interpreted not as prescriptive dogma but as testable hypotheses, bridging the gap between evidence and patient experience and fostering a more integrated, biopsychosocial model of IBS care.

## Figures and Tables

**Figure 1 nutrients-17-02496-f001:**
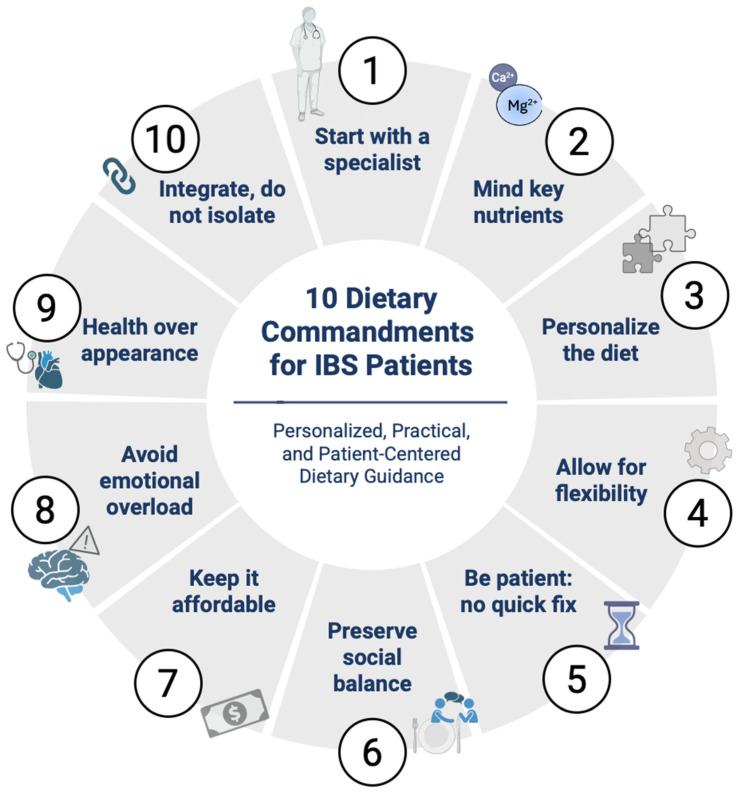
10 Dietary Commandments for IBS.

**Table 1 nutrients-17-02496-t001:** The table provides a classification of foods based on their FODMAP content, divided into three main categories: allowed foods, foods to be consumed in moderation, and foods to avoid.

Category	Allowed Foods (Low FODMAP Content)	Foods to Limit (Moderate FODMAP Content)	Foods to Avoid (High FODMAP Content)
Vegetables	Zucchini, carrots, tomatoes, spinach, lettuce, cucumbers, eggplant, bell peppers, green beans, radishes	Beets, avocado (≤30 g of avocado pulp), champignon mushrooms, pumpkin, broccoli (only florets), Brussels sprouts (small portions), sweet corn, fennel, sweet potatoes (in small amounts), frozen peas	Onion, garlic, cauliflower, leeks, asparagus, savoy cabbage, spring onion, portobello mushrooms, broccoli stems, artichokes
Fruits	Unripe bananas, strawberries, blueberries, raspberries, kiwi, oranges, mandarins, grapes, dragon fruit, cantaloupe melon	Dried plums (small amounts), apricots (½ fruit), figs (1–2 fruits), papaya, grapefruit, raisins, nectarines, lime (limited quantities), currants, pineapple (in moderation)	Apples, pears, mango, watermelon, cherries, peaches, nectarines, plums, lychee, mixed berries
Cereals and derivatives	Rice, oats, quinoa, corn, polenta, rice pasta, gluten-free bread (made with FODMAP-friendly flours), tapioca, buckwheat, millet	Whole rye bread (small portions), pearled spelt, hulled barley, couscous (small amounts), crackers with small quantities of wheat, lentil pasta (small amounts), whole grain cereals, oat flour, corn cakes	Wheat, rye, barley (large amounts), white bread, regular pasta, standard cookies, cereals with honey/fructose, crackers, soft wheat flour, couscous, industrial baked goods
Dairy products	Lactose-free milk, aged cheeses, clarified butter, lactose-free yogurt, lactose-free cream, brie cheese (small amounts), mozzarella (limited portion), cheddar, feta, plant-based milks, kefir	Greek yogurt (moderate amounts), diluted cow’s milk, soft cheeses, cooking cream (small amounts), lactose-free ricotta, plant-based cream (watch for additives), soy yogurt, oat milk (small amounts), blue cheeses, fermented milk	Cow’s milk, goat’s milk, regular yogurt, full cream, cream cheese, mascarpone, fresh cheeses (e.g., primo sale), condensed milk, sheep’s milk, industrial ice cream
Proteins	Beef, chicken, turkey, fish (tuna, salmon, cod), eggs, tofu, tempeh, cured ham, untreated bacon, bresaola, seafood	Cured meats with additives (nitrates, dextrose), marinated meat, soft tofu, processed burgers, canned meat, flavored canned tuna, hot dogs, sausage (small amounts), pre-breaded packaged meat, processed shellfish	Lentils, beans, chickpeas, soy, fava beans, dried peas, vegetable burgers with fructans, unfixed tofu, protein-enriched foods with polyols, miso, industrial legume soups
Beverages	Water, green tea, black tea, espresso coffee, herbal teas (mint, ginger, chamomile), almond milk (unsweetened), rice milk, diluted fruit juices (from allowed fruits), coconut water (small amounts), broth	Light beer (small amounts), dry red or white wine, plant-based drinks, coffee with lactose-free milk, 100% fruit juices, matcha tea, kombucha, sugar-free energy drinks (in moderation), flavored water (natural), mixed fruit smoothies (watch ingredients)	Carbonated drinks with sweeteners (e.g., sorbitol, mannitol), apple, pear or mango juices, dark beer, cow’s milk, sweet liqueurs, sweetened alcoholic beverages, soy-based drinks, industrial milkshakes, energy drinks with polyol sugars
Sweeteners	White sugar, brown sugar, maple syrup, glucose, pure stevia, coconut sugar (in moderation), rice syrup, corn syrup, blackstrap molasses, erythritol (moderate amounts)	Honey (≤1 teaspoon), invert sugar, maltose, isomalt (moderation), agave syrup (very limited), stevia with erythritol, mixed sweeteners, candies with maltodextrin, sugar-free gums (without polyols), dark chocolate (small amounts)	Sorbitol, mannitol, xylitol, maltitol, isomalt (in excess), honey in large quantities, agave syrup, diabetic sweeteners, chewing gums with polyols, sugar-free candies, filled chocolates

**Table 2 nutrients-17-02496-t002:** Details of Mediterranean diet foods and their consumption frequencies [45,46].

Food Category	Frequency/Consumption	Notes/Details
Whole grains and legumes	Daily	Fundamental basis of the diet; prefer whole, seasonal, and locally sourced products
Fresh vegetables	Daily	Consume a wide variety; emphasize seasonal and local choices
Fresh fruit	Daily	Used as dessert in main meals; select seasonal varieties
Hydration and physical activity	Daily	Water as the primary beverage; engage in regular physical activity
Sweets	Occasional	Prepared with dried fruit, olive oil, and honey; reserved for celebrations
Fish	Moderate	Mainly oily fish rich in omega-3 fatty acids
Poultry and eggs	Weekly	Consume in moderation as alternatives to red and processed meats
Fat sources	Daily	Mainly extra virgin olive oil, nuts, and seeds
Dairy products	Reduced	Prefer fermented products like local yogurt and cheeses; limited use of butter and cream
Red and processed meat	3–4 times per month	Limit intake; opt for small portions
Wine	Occasional	Not recommended, but if culturally relevant, limit to small amounts during meals and only in appropriate social contexts

## Data Availability

The original contributions presented in the study are included in the article, further inquiries can be directed to the corresponding author.

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
