# Peer review of "The Ten Dietary Commandments for Patients with Irritable Bowel Syndrome: A Narrative Review with Pragmatic Indications"

_nutrients, 2025, doi:10.3390/nu17152496_

Round 1
Reviewer 1 Report
Comments and Suggestions for Authors
Comments to the Authors of manuscript number nutrients-3769751 entitled “The Ten Dietary Commandments for Patients with Irritable Bowel Syndrome: a Narrative Review with Pragmatic Indications”
- Abstract, 28: “Its multifactorial pathophysiology involves gut microbiota dysbiosis, visceral hypersensitivity, disordered motility, low-grade inflammation, and psychosocial factors.” Although dysbiosis and hypersensitivity are well documented, the role of “low-grade inflammation” in IBS remains controversial and requires stronger support in meta-analyses (e.g., Ford et al. 2020 does not unequivocally support inflammation in IBS). It lacks precise citations of studies demonstrating the clinical significance of this phenomenon.
- Abstract, 33–34: “The Mediterranean diet, due to its anti-inflammatory and prebiotic profile, offers a more sustainable, microbiota-friendly option.” The authors cite the anti-inflammatory benefits of MD, but in the context of IBS, there are no direct RCTs confirming the advantages of this diet over other dietary interventions. The Kasti et al. 2022 study they cite is a protocol for combining MD with low-FODMAP, not a pure RCT of MD vs. control in patients with IBS. A more critical discussion of the quality of the evidence is necessary.
- Introduction, lines 52–54: "…with a prevalence ranging from 3.8% to 12%… Specifically, the global prevalence of IBS has been estimated at 4.1% based on the Rome IV… compared to 9.2% using the Rome III criteria [5,6]." The ranges provided (3.8–12%) are too broad and include both clinical and survey populations. The references [5,6] are primarily diagnostic guidelines, not reliable epidemiological reviews. Please reference a specific population meta-analysis or specify the studies supporting this.
- Table 1, line 284: In the "Fruits – Foods to Limit" section, the authors list avocados ≤ 30 g as a moderate source of FODMAPs. According to Monash University, avocados contain primarily sorbitol and the allowable serving size is 30 g, but this limit should be described more precisely (e.g., 30 g of flesh, not the whole fruit) and the source osmolality value should be provided. There is no explanation of where the authors derived their serving limits.
- Section 2.1.3 IgG-guided elimination diet, lines 426–428: "This theory is highly debated… because food-related IgG antibodies can also be found in healthy individuals [59]." The authors rightly point out the controversies, but they do not cite a Cochrane review or meta-analysis (Atkinson et al. 2004 is a single RCT from 20 years ago). Criticism of the quality of the studies and the lack of standardization of IgG tests are worth adding.
- "Ten Commandments for IBS," lines 41–47: The 10 "commandments" proposal is attractive as a clinical tool, but the authors do not present any studies validating the effectiveness of this algorithm. There is a lack of data on improved adherence or symptom outcomes after using this model in prospective studies. Therefore, the conclusions from lines 41–47 are largely unscientific and require at least a pilot evaluation.
- Methodology, lines 126–133: "A comprehensive literature search was carried out … Google Scholar … A total of 188 English-language studies were selected…" There is no description of the selection criteria (inclusion/exclusion), the selection process (PRISMA), or the risk of bias assessment. It is necessary to clarify what search algorithms and methodological criteria were used.
Author Response
Comments 1: Abstract, 28: “Its multifactorial pathophysiology involves gut microbiota dysbiosis, visceral hypersensitivity, disordered motility, low-grade inflammation, and psychosocial factors.” Although dysbiosis and hypersensitivity are well documented, the role of “low-grade inflammation” in IBS remains controversial and requires stronger support in meta-analyses (e.g., Ford et al. 2020 does not unequivocally support inflammation in IBS). It lacks precise citations of studies demonstrating the clinical significance of this phenomenon.
Response 1: We thank the reviewer for their insightful comment. Indeed, as rightly pointed out, the role of low-grade inflammation in IBS remains controversial and is not sufficiently supported by robust meta-analyses. To ensure consistency between the abstract and the main manuscript content, we have removed the mention of “low-grade inflammation” from the abstract. Within the main text, we have focused on the more established mechanisms, such as gut microbiota dysbiosis, visceral hypersensitivity, motility alterations, and psychosocial factors, in accordance with current evidence and the cited references. We appreciate the reviewer’s suggestion, which has contributed to enhancing the clarity and accuracy of our work.
Comments 2: Abstract, 33–34: “The Mediterranean diet, due to its anti-inflammatory and prebiotic profile, offers a more sustainable, microbiota-friendly option.” The authors cite the anti-inflammatory benefits of MD, but in the context of IBS, there are no direct RCTs confirming the advantages of this diet over other dietary interventions. The Kasti et al. 2022 study they cite is a protocol for combining MD with low-FODMAP, not a pure RCT of MD vs. control in patients with IBS. A more critical discussion of the quality of the evidence is necessary.
Response 2: We thank the reviewer for their comment, which gives us the opportunity to further clarify our position. Indeed, as rightly noted, there are no high-quality randomized controlled trials definitively demonstrating the superiority of the Mediterranean diet alone in managing IBS compared to other dietary interventions. In our manuscript, however, we have aimed to maintain a balanced tone by highlighting both the potential benefits of the Mediterranean diet—particularly regarding microbiota modulation and systemic inflammation—and its specific limitations in the context of IBS, including the adverse effects of certain FODMAP-rich foods (lines 32 - 35, 353 - 355). We also emphasized the importance of a personalized approach, supervised by healthcare professionals, and discussed the possibility of combining Mediterranean diet principles with those of the low-FODMAP diet, as proposed by Kasti et al. (lines 393 – 397). We believe our text does not present the Mediterranean diet as a one-size-fits-all solution but rather as a potentially sustainable dietary foundation to be carefully adapted according to individual patient characteristics.
Comments 3: Introduction, lines 52–54: "…with a prevalence ranging from 3.8% to 12%… Specifically, the global prevalence of IBS has been estimated at 4.1% based on the Rome IV… compared to 9.2% using the Rome III criteria [5,6]." The ranges provided (3.8–12%) are too broad and include both clinical and survey populations. The references [5,6] are primarily diagnostic guidelines, not reliable epidemiological reviews. Please reference a specific population meta-analysis or specify the studies supporting this.
Response 3: We thank the reviewer for their comment. However, we wish to clarify that the prevalence range reported (3.8%–12%) reflects what is documented in the international scientific literature, considering the variability of diagnostic criteria used (Rome III vs. Rome IV) and the geographic diversity of the studied populations. References [5] and [6] are cited solely to contextualize the evolution of diagnostic criteria, whereas the epidemiological data are supported by a large-scale meta-analysis conducted across 38 countries involving approximately 400,000 individuals (Oka et al., Lancet Gastroenterol Hepatol, 2020 – reference [7]), which provides precise estimates of the global prevalence of IBS. Therefore, we consider the text to report accurate and well-documented information.
Comments 4: Table 1, line 284: In the "Fruits – Foods to Limit" section, the authors list avocados ≤ 30 g as a moderate source of FODMAPs. According to Monash University, avocados contain primarily sorbitol and the allowable serving size is 30 g, but this limit should be described more precisely (e.g., 30 g of flesh, not the whole fruit) and the source osmolality value should be provided. There is no explanation of where the authors derived their serving limits.
Response 4: We thank the reviewer for the detailed comment. In Table 1, the indication of avocado as a “moderate FODMAP content” food with a serving limit of ≤30 g specifically refers to the edible flesh, not the whole fruit. We agree that this distinction should have been made clearer, and we have accordingly revised the text to specify “≤30 g of avocado flesh” (line 296). This serving recommendation is based on data published by the Monash University group, which we have cited in our reference list (Reference 19: Gibson PR et al., J Gastroenterol Hepatol. 2010). These authors originally established the FODMAP classification system and portion size guidelines based on their analytical measurements and clinical thresholds. Regarding the osmolality value, we acknowledge the reviewer’s request; however, such detailed physicochemical data are not always available for all foods listed in the literature and were not the primary focus of this summary table. Our intention was to provide clinicians and patients with practical, evidence-based guidance regarding portion sizes founded on validated FODMAP content thresholds. We believe this maintains the clinical utility of the table. We thank the reviewer again for highlighting this opportunity for clarification.
Comments 5: Section 2.1.3 IgG-guided elimination diet, lines 426–428: "This theory is highly debated… because food-related IgG antibodies can also be found in healthy individuals [59]." The authors rightly point out the controversies, but they do not cite a Cochrane review or meta-analysis (Atkinson et al. 2004 is a single RCT from 20 years ago). Criticism of the quality of the studies and the lack of standardization of IgG tests are worth adding.
Response 5: Thank you for the comment. We have accepted the suggestion and added a concluding paragraph to the section on the IgG-guided elimination diet, clearly highlighting the limitations of the current literature (lines 484 – 489). Specifically, we emphasize the absence of meta-analyses, the lack of standardization of IgG testing, and the scarcity of long-term data. This addition strengthens the critical framework regarding the use of these tests in clinical practice.
Comments 6: "Ten Commandments for IBS," lines 41–47: The 10 "commandments" proposal is attractive as a clinical tool, but the authors do not present any studies validating the effectiveness of this algorithm. There is a lack of data on improved adherence or symptom outcomes after using this model in prospective studies. Therefore, the conclusions from lines 41–47 are largely unscientific and require at least a pilot evaluation.
Response 6: Thank you for your valuable comment. In the Conclusions, we aimed to highlight the practical potential of implementing the “Ten Dietary Commandments for IBS” in clinical practice (lines 1147 – 1151). Although these represent an innovative approach developed from clinical experience, we acknowledge that they have not yet undergone systematic validation through prospective or implementation studies. Nevertheless, we believe these principles can offer a useful and easily applicable guide for dietary management of IBS, and our future goal is to evaluate their real-world effectiveness.
Comments 7: Methodology, lines 126–133: "A comprehensive literature search was carried out … Google Scholar … A total of 188 English-language studies were selected…" There is no description of the selection criteria (inclusion/exclusion), the selection process (PRISMA), or the risk of bias assessment. It is necessary to clarify what search algorithms and methodological criteria were used.
Response 7: We thank the reviewers for their thorough and constructive comments. We acknowledge the importance of providing a clear and transparent description of the search strategy, study selection process, inclusion/exclusion criteria, and clarification regarding any risk of bias assessment. In response to the reviewers' observations, we have incorporated a comprehensive description of the literature search strategy (lines 124 – 146). This search was conducted across multiple databases (PubMed, Scopus, Web of Science, Google Scholar, and Cochrane Library), using Boolean operators and an exhaustive set of search terms related to irritable bowel syndrome and dietary interventions. A total of 247 articles were initially identified; of these, 188 English-language, peer-reviewed studies were included based on explicit inclusion criteria such as study design (RCTs, meta-analyses, systematic reviews, clinical guidelines), clinical relevance, and methodological quality. Non-peer-reviewed studies, publications in languages other than English, and non-relevant articles were excluded. Select clinically relevant case reports were included when deemed illustrative or contextually informative. Study selection was performed independently by four reviewers to minimize selection bias; any disagreements were resolved through discussion and consensus. Although a formal PRISMA protocol was not followed, a systematic, transparent, and reproducible methodology was adopted, consistent with the narrative nature of this review. Regarding risk of bias assessment, given the narrative design and heterogeneity of included studies, a formal evaluation was not conducted. Nevertheless, study quality was implicitly considered through the selection criteria emphasizing peer-reviewed status and clinical relevance. We believe that these methodological clarifications and additions adequately address the reviewers’ concerns, enhancing the transparency and rigor of the work without compromising the pragmatic and integrative approach of our review.
Reviewer 2 Report
Comments and Suggestions for Authors
This manuscript reviews dietary changes for irritable bowel syndrome (IBS) and introduces a new idea called “Ten Dietary Commandments” to help with clinical practice. It provides a clear comparison of different dietary strategies. The paper effectively connects the reasons behind these strategies, their clinical results, the effects on gut bacteria, and the role of mental and social factors. This makes the information relevant for clinical use.
This is an excellent and clinically valuable review that may serve as a reference point for both specialists and primary care providers managing IBS.
Main comments:
IgG-based elimination diets should be approached carefully. It is important to highlight their controversial nature more clearly to avoid any misunderstanding by doctors or patients.
Although it is briefly mentioned in the text, the authors may consider expanding on how cost, access, and cultural dietary patterns determine the feasibility of implementing these dietary strategies on a global scale.
The manuscript wisely endorses professional dietary guidance. In this context, a brief discussion on how multidisciplinary care (e.g., psychologists or exercise physiologists) could improve outcomes would enrich the paper.
Author Response
Comments 1: IgG-based elimination diets should be approached carefully. It is important to highlight their controversial nature more clearly to avoid any misunderstanding by doctors or patients.
Response 1: Thank you for your valuable comment. We have accepted the suggestion and added a concluding paragraph in the section on IgG-guided elimination diets that clearly highlights the limitations of the current literature (lines 484 – 489). Specifically, we emphasize the lack of meta-analyses, the absence of standardization in IgG testing, and the scarcity of long-term data. This addition strengthens the critical framework regarding the use of these tests in clinical practice.
Comments 2: Although it is briefly mentioned in the text, the authors may consider expanding on how cost, access, and cultural dietary patterns determine the feasibility of implementing these dietary strategies on a global scale.
Response 2: Thank you for your valuable comment. In the manuscript (lines 93 – 123), we introduced the socio-cultural dimensions of food, along with the factors that influence the adoption and sustainability of dietary strategies. In the conclusions (lines 1147 – 1151), we emphasized the practical potential for translating the “10 Dietary Commandments for IBS” into clinical practice, which we believe may serve as a useful and easily applicable guide for the dietary management of IBS. Our future goal is to evaluate their real-world effectiveness. However, we acknowledge that, although these recommendations represent an innovative approach developed from clinical experience, they have not yet undergone systematic validation through prospective or implementation studies. Therefore, we are currently unable to accurately define the costs and feasibility of access or implementation on a global scale, considering the variability in cultural dietary models.
Comments 3: The manuscript wisely endorses professional dietary guidance. In this context, a brief discussion on how multidisciplinary care (e.g., psychologists or exercise physiologists) could improve outcomes would enrich the paper.
Response 3: Thank you for the valuable comment. The manuscript already includes a discussion on the importance of a multidisciplinary approach in the care of patients with IBS, highlighting the complementary roles of gastroenterologists, dietitians, nutritionists, psychiatrists, behavioral therapists, as well as other specialists such as pelvic physiotherapists and gut-focused hypnotherapists (section 2.6). This integrated approach has been described as superior to gastroenterology care alone, improving symptoms, psychological well-being, quality of life, and reducing healthcare costs.
Reviewer 3 Report
Comments and Suggestions for Authors
The article "The Ten Dietary Commandments for Patients with Irritable Bowel Syndrome: A Narrative Review with Pragmatic Indications" by Nicola Siragusa et al. offers valuable insights, though several aspects merit further consideration:
- While the authors describe these diets as "controversial," they do not sufficiently address the lack of robust evidence supporting their use in IBS, which may lead to misinterpretation by clinicians and patients.
- Although highlighted as a sustainable option, the review overlooks critical factors such as cost, cultural adaptability, and feasibility for low-income populations—key considerations for real-world implementation.
- Despite mentioning psychosocial influences, the review does not thoroughly examine the interplay between stress, eating behaviors and disordered eating patterns, which are prevalent among IBS patients.
- While a simplified model (the "10 Commandments") is practical, it risks reducing IBS management to a rigid checklist without accounting for individual variability in symptom triggers, comorbidities (e.g., SIBO, GERD), or evolving scientific evidence.
- The concern about microbiota diversity is valid, but the review could elaborate on mitigation strategies, such as phased reintroduction or synbiotics, to address potential risks.
- These are briefly noted in the context of gluten-free diets but are not explored in relation to other dietary interventions, despite their significant influence on IBS trial outcomes.
- A summary table contrasting the pros and cons of each diet (efficacy, adherence difficulty, microbiota effects) could enhance the review’s clinical utility.
Author Response
Comments 1: While the authors describe these diets as "controversial," they do not sufficiently address the lack of robust evidence supporting their use in IBS, which may lead to misinterpretation by clinicians and patients.
Response 1: Thank you for the valuable comment. However, we consider it important to clarify that the main objective of our review is precisely to critically analyze, in a balanced and thorough manner, the primary dietary strategies used in the management of IBS. We emphasize not only the supporting evidence but also—and especially—the scientific limitations, potential risks, and ongoing controversies in the literature. As highlighted in Sections 2.1.1 through 2.1.6, none of these diets are presented as one-size-fits-all or universally applicable solutions. On the contrary, their use is emphasized as being appropriate only for specific subgroups of patients and always under specialist supervision, in order to avoid misinterpretation by both patients and healthcare providers. The ultimate purpose of the work, summarized in the “Ten Dietary Commandments for IBS,” is to provide a pragmatic clinical tool aimed at preventing rigid, reductionist, or potentially harmful approaches, while promoting personalized, integrated, and informed management based on a balance between scientific evidence and clinical common sense.
Comments 2: Although highlighted as a sustainable option, the review overlooks critical factors such as cost, cultural adaptability, and feasibility for low-income populations—key considerations for real-world implementation.
Response 2: Thank you for the valuable comment. In the conclusions (lines 1147 – 1151), we highlighted the concrete possibility of translating the “10 Dietary Commandments for IBS” into clinical practice, which we believe can serve as a useful and easily applicable guide in the dietary management of IBS. Our goal, in the near future, is to evaluate their effectiveness in real-world settings. However, we acknowledge that although they represent an innovative approach developed from clinical experience, they have not yet undergone systematic validation through prospective or implementation studies. Therefore, we are not yet able to precisely outline the costs, cultural adaptability, and feasibility for low-income populations.
Comments 3: Despite mentioning psychosocial influences, the review does not thoroughly examine the interplay between stress, eating behaviors and disordered eating patterns, which are prevalent among IBS patients.
Response 3: Thank you for the comment. We fully agree on the importance of the interplay between psychosocial factors, stress, and eating behaviors in patients with IBS. As noted in the introductory section of our review, we acknowledge the multifactorial nature of the syndrome, including the complex gut-brain axis interactions and psycho-emotional influences. Furthermore, in our dietary “Ten Commandments” (pag. 26), particularly point 8, we explicitly address the risk of developing a dysfunctional and obsessive relationship with food, which can undermine the effectiveness of nutritional interventions and negatively impact quality of life. We also reiterated the need for an integrated biopsychosocial approach to IBS treatment in the concluding section. However, since this is a narrative review focused primarily on dietary strategies, we chose not to delve exhaustively into all psychological and behavioral aspects to maintain the manuscript’s focus, coherence, and accessibility. We consider the integration of dietary and psychological aspects a crucial direction for future studies specifically dedicated to this field.
Comments 4: While a simplified model (the "10 Commandments") is practical, it risks reducing IBS management to a rigid checklist without accounting for individual variability in symptom triggers, comorbidities (e.g., SIBO, GERD), or evolving scientific evidence.
Response 4: Thank you for the insightful comment. We understand the concern that a simplified model like the "10 Dietary Commandments for IBS" might be perceived as a rigid, prescriptive checklist. However, the explicit aim of our commandments is quite the opposite: to provide practical and flexible principles that help both patients and clinicians move beyond the rigidity and strictness often associated with many IBS dietary strategies. These commandments are rooted in clinical experience and the understanding that there is no “one-size-fits-all” diet; rather, every nutritional intervention should be individualized, dynamic, and sustainable over time. Furthermore, in the manuscript, we explicitly emphasize that the commandments are not to be taken as dogma but as guiding tools intended to promote dietary awareness, improve adherence, and reduce risks associated with overly rigid interpretations of diets.
Comments 5: The concern about microbiota diversity is valid, but the review could elaborate on mitigation strategies, such as phased reintroduction or synbiotics, to address potential risks.
Response 5: Thank you for raising a key issue in the dietary management of IBS, namely the impact of dietary restrictions on gut microbiota diversity. In our manuscript (section 2.1.1), we have indeed highlighted this risk in relation to the low-FODMAP diet, emphasizing that reduced microbial diversity is one of the main concerns of this protocol, especially when prolonged over time. In this regard, we discuss the importance of the gradual reintroduction phase (Phase 2), which is a fundamental strategy to mitigate the risk of long-term side effects while simultaneously improving dietary variety. Additionally, in section 2.5 we mention the use of probiotics —both in the main text and in the final dietary commandments—as potential complementary supports to modulate the microbiota during more restrictive dietary interventions.
Comments 6: These are briefly noted in the context of gluten-free diets but are not explored in relation to other dietary interventions, despite their significant influence on IBS trial outcomes.
Response 6: Thank you for the reviewer’s insightful observation regarding the importance of contextualizing aspects such as food reintroduction, dietary flexibility, and the role of the microbiota in influencing dietary outcomes in IBS patients. However, we would like to emphasize that these elements are not discussed solely in relation to the gluten-free diet but rather represent a common thread running through multiple sections of our review. Specifically: in the discussion of the low-FODMAP diet (section 2.1.1), we highlight the need for a gradual and structured reintroduction phase to promote microbial diversity recovery and improve individual tolerance; in the section 2.1.2 on the Mediterranean diet, we address the limitations related to FODMAP content and the necessity for personalized adaptations; within our “Dietary Commandments,” several principles underscore flexibility, personalization, and long-term sustainability to avoid rigid approaches and foster a holistic, adaptive management of the condition; finally, we also mention (section 2.5) the potential use of probiotics as supportive tools where appropriate.
Comments 7: A summary table contrasting the pros and cons of each diet (efficacy, adherence difficulty, microbiota effects) could enhance the review’s clinical utility.
Response 7: Thank you to the reviewer for the valuable suggestion. However, we would like to point out that the manuscript already includes a summary table (Figure 1, on page 5) that compares the main dietary strategies for IBS. This table highlights the strengths, limitations, adherence challenges, microbiota impact, and clinical rationale for each dietary approach discussed. We believe that this existing table fully addresses the reviewer’s request and is intended precisely to enhance the clinical utility and readability of the review for the reader.
Reviewer 4 Report
Comments and Suggestions for Authors
This manuscript provides a narrative review of current dietary strategies used in patients with irritable bowel syndrome (IBS), concluding with a proposal for "10 dietary commandments." The work is comprehensive, well-documented, and readable. However, despite its practicality and narrative style, it contains some shortcomings that should be addressed before publication. The main issues include unclear methodology, selective citations, the risk of overinterpretation of data, and incomplete integration of practical aspects with scientific evidence.
Lack of a clearly defined narrative review methodology: The authors declare a literature search (PubMed, Scopus, etc.), but do not provide details regarding the number of included articles per diet, the selection strategy, or the quality assessment process. Please supplement this section with a search scheme and at least simplified inclusion/exclusion criteria.
An overly selective approach to the source data – the review focuses almost exclusively on the positive effects of diets, omitting significant limitations, such as research showing the lack of effectiveness of a low-FODMAP diet or the possible negative effects of a gluten-free diet in people without celiac disease.
The "10 Dietary Commandments" as a narrative structure – an interesting idea, but it is unclear on what basis they were formulated. Are they the result of personal clinical observations, a survey, or expert consensus? A justification of their source and possible limitations is necessary.
A graphical diagram of the relationships between diets, microbiota, and IBS symptoms is missing – although a general diagram appears on page 25 (Figure 2), it would be worthwhile to supplement the text with at least one graph illustrating the effects of the main diets on the microbiome, cytokines, SCFAs, etc. is problematic. Although the authors acknowledge the controversy, they overemphasize their potential benefits while lacking sufficient clinical data. In this format, it can lead readers to erroneous conclusions.
Poor integration of clinical data with pathophysiology – the authors mention neurotransmitters, SCFAs, GALT, HPA, etc., but do not create a coherent, integrated narrative of how specific diets affect these pathways in patients with IBS.
The introductory section (pp. 2–4) is too extensive – consider shortening or moving some cultural and social references to an appendix; their current form does not add substantive value to the review.
There is no mention of the risk of nutritional deficiencies with elimination diets – it would be worthwhile to expand on potential deficiencies (Ca, Fe, Zn, B12), especially with long-term use of lactose-free and low-FODMAP diets.
On page 5 (Figure 1): the table should standardize the terminology regarding "tolerability" and "scientific support" – some columns are subjective, others are data-driven.
Some of the data regarding gut microbiota is general – there is no specific information regarding changes in Bifidobacterium, Akkermansia, Faecalibacterium, etc.
On page 20, the authors mention the risk of orthorexia – it would be helpful to provide specific numerical data (e.g., screening rates).
Please supplement the citations in the section on probiotics (p. 22) with meta-analyses.
Author Response
Comments 1: This manuscript provides a narrative review of current dietary strategies used in patients with irritable bowel syndrome (IBS), concluding with a proposal for "10 dietary commandments." The work is comprehensive, well-documented, and readable. However, despite its practicality and narrative style, it contains some shortcomings that should be addressed before publication. The main issues include unclear methodology, selective citations, the risk of overinterpretation of data, and incomplete integration of practical aspects with scientific evidence.
Response 1: We thank the reviewers for their thorough and constructive comments. We acknowledge the importance of providing a clear and transparent description of the search strategy, study selection process, inclusion and exclusion criteria, and clarification regarding any risk of bias assessment. In response to the reviewers’ observations, we have incorporated a comprehensive description of the literature search strategy (lines 124 – 146). This search was conducted across multiple databases (PubMed, Scopus, Web of Science, Google Scholar, and Cochrane Library), using Boolean operators and an exhaustive set of search terms related to irritable bowel syndrome and dietary interventions. A total of 247 articles were initially identified; of these, 188 English-language, peer-reviewed studies were included based on explicit inclusion criteria such as study design (RCTs, meta-analyses, systematic reviews, clinical guidelines), clinical relevance, and methodological quality. Non-peer-reviewed studies, publications in languages other than English, and non-relevant articles were excluded. Select clinically relevant case reports were included when deemed illustrative or contextually informative. Study selection was performed independently by four reviewers to minimize selection bias; any disagreements were resolved through discussion and consensus. Although a formal PRISMA protocol was not followed, a systematic, transparent, and reproducible methodology was adopted, consistent with the narrative nature of this review. Regarding risk of bias assessment, given the narrative design and heterogeneity of included studies, a formal evaluation was not conducted. Nevertheless, study quality was implicitly considered through the selection criteria emphasizing peer-reviewed status and clinical relevance. Finally, regarding the integration of practical aspects with scientific evidence, we would like to emphasize that the “10 Dietary Commandments for IBS” represent a pragmatic model mainly derived from clinical experience and inspired by principles from other dietary strategies. This model is designed to guide personalized and flexible management. We acknowledge the need for future validation through prospective studies, but we believe this approach provides an effective balance between evidence and clinical applicability. We trust that these methodological clarifications and additions adequately address the reviewers’ concerns, enhancing the transparency and rigor of the work without compromising the pragmatic and integrative approach of our review.
Comments 2: Lack of a clearly defined narrative review methodology: The authors declare a literature search (PubMed, Scopus, etc.), but do not provide details regarding the number of included articles per diet, the selection strategy, or the quality assessment process. Please supplement this section with a search scheme and at least simplified inclusion/exclusion criteria.
Response 2: We thank the reviewer for this insightful observation, which allowed us to clarify the methodology adopted in our narrative review. In response, we have provided a more transparent description of the literature search strategy, the study selection process, and the inclusion/exclusion criteria lines 124 – 146). The search was conducted across five major databases (PubMed, Scopus, Web of Science, Google Scholar, and the Cochrane Library) up to July 2025, using Boolean operators to combine key terms related to irritable bowel syndrome and major dietary strategies (e.g., “IBS,” “low-FODMAP,” “gluten-free,” “Mediterranean diet”). A total of 247 articles were initially identified. After removal of duplicates and application of predefined criteria, 188 English-language, peer-reviewed articles were included, based on clinical relevance and methodological quality. Inclusion criteria encompassed randomized controlled trials (RCTs), meta-analyses, systematic reviews, clinical guidelines, and, to a lesser extent, narrative reviews and illustrative case reports. Studies not focused on IBS, non-peer-reviewed publications, and articles in languages other than English were excluded. Study selection was conducted independently by four reviewers, and discrepancies were resolved by discussion and consensus. Although a formal PRISMA protocol was not followed—given the narrative nature of the review—the adopted methodology was systematic, reproducible, and appropriate for the aims of the paper. A formal risk of bias assessment was not performed, but study quality was implicitly considered during the selection process. We believe these revisions adequately address the reviewer’s concern, improving the transparency and methodological rigor of our manuscript.
Comments 3: An overly selective approach to the source data – the review focuses almost exclusively on the positive effects of diets, omitting significant limitations, such as research showing the lack of effectiveness of a low-FODMAP diet or the possible negative effects of a gluten-free diet in people without celiac disease.
Response 3: Thank you for the valuable comment. Although striking a balance between presenting benefits and limitations was not straightforward, we aimed to provide an analysis that addresses not only the positive effects of the diets but also their main limitations and potential risks. In particular, we discussed the evidence questioning the long-term effectiveness of the low-FODMAP diet (section 2.1.1), as well as the possible negative consequences of a gluten-free diet in non-celiac individuals (section 2.1.6), such as alterations in microbiota composition and challenges with adherence. This critical and balanced approach underpins the formulation of our “Ten Dietary Commandments for IBS management,” which serves as a practical synthesis based on evidence, designed to guide personalized, sustainable, and informed dietary choices. We believe this method strengthens the scientific rigor of our review and enhances its clinical relevance.
Comments 4: The "10 Dietary Commandments" as a narrative structure – an interesting idea, but it is unclear on what basis they were formulated. Are they the result of personal clinical observations, a survey, or expert consensus? A justification of their source and possible limitations is necessary.
Response 4: Thank you for the valuable comment. In the Conclusions (lines 1147 – 1151), we emphasized the practical applicability of the “Ten Dietary Commandments for IBS.” Although they represent an innovative approach developed from clinical experience, we acknowledge that they have not yet undergone systematic validation through prospective or implementation studies. Nonetheless, we believe these principles offer a useful and easily applicable guide for dietary management of IBS. Our future aim is to evaluate their effectiveness in real-world clinical settings.
Comments 5: A graphical diagram of the relationships between diets, microbiota, and IBS symptoms is missing – although a general diagram appears on page 25 (Figure 2), it would be worthwhile to supplement the text with at least one graph illustrating the effects of the main diets on the microbiome, cytokines, SCFAs, etc. is problematic. Although the authors acknowledge the controversy, they overemphasize their potential benefits while lacking sufficient clinical data. In this format, it can lead readers to erroneous conclusions.
Response 5: Thank you for the valuable comment. We agree that graphical representations are valuable in review articles; however, we would like to clarify that the focus of our manuscript is on the critical analysis of the main diets used in IBS management, with particular attention to their strengths and limitations, rather than on the specific effects of these diets on biomolecular components or the gut microbiota. A detailed and systematic discussion of microbiome alterations, cytokines, or metabolites such as SCFAs would go beyond the scope of this review and risk excessively broadening the manuscript’s focus. For these reasons, we chose not to include a diagram exploring these aspects in detail, while acknowledging their importance for future, more targeted studies.
Comments 6: Poor integration of clinical data with pathophysiology – the authors mention neurotransmitters, SCFAs, GALT, HPA, etc., but do not create a coherent, integrated narrative of how specific diets affect these pathways in patients with IBS.
Response 6: Thank you for the valuable comment. We are fully aware of the importance of biomolecular and pathophysiological mechanisms in understanding irritable bowel syndrome. However, we felt that delving deeply into these aspects might have made the manuscript overly complex, which is already quite detailed and primarily focused on a critical description of the main dietary strategies, highlighting their strengths and weaknesses. Our goal was to provide a clinically oriented and easily accessible synthesis, leaving more in-depth pathophysiological discussions to studies specifically dedicated to these mechanisms.
Comments 7: The introductory section (pp. 2–4) is too extensive – consider shortening or moving some cultural and social references to an appendix; their current form does not add substantive value to the review.
Response 7: Thank you for the valuable comment. We considered it important to maintain a comprehensive introduction in order to provide a multidimensional framework for dietary management in IBS, including relevant cultural and social aspects. This context supports a more complete understanding of the complex interactions between diet, behavior, and symptomatology, facilitating the interpretation of the clinical and therapeutic considerations presented in the review. We believe this approach is beneficial for an interdisciplinary audience and helps to appropriately contextualize the clinical evidence discussed.
Comments 8: There is no mention of the risk of nutritional deficiencies with elimination diets – it would be worthwhile to expand on potential deficiencies (Ca, Fe, Zn, B12), especially with long-term use of lactose-free and low-FODMAP diets.
Response 8: Thank you for the valuable comment. We have integrated the manuscript with a dedicated section on the potential nutritional risks associated with elimination diets. In paragraph 2.4, "Macro and Micronutrients," we discuss possible deficiencies in macro- and micronutrients (including calcium, iron, zinc, vitamin B12, vitamin D, magnesium, and selenium) frequently observed in patients with IBS, emphasizing their impact on nutritional status and the need for careful monitoring and targeted supplementation, especially in cases of prolonged dietary restrictions.
Comments 9: On page 5 (Figure 1): the table should standardize the terminology regarding "tolerability" and "scientific support" – some columns are subjective, others are data-driven.
Response 9: Thank you for the valuable suggestion regarding the standardization of terminology related to “tolerability” and “scientific support” in Table 1. We would like to clarify that the assessments presented in the table are based on a narrative synthesis of the currently available literature analyzed in our article, which includes randomized controlled trials, meta-analyses, and international guidelines. At present, there is no universally accepted validated tool to assign comparative quantitative scores to different dietary approaches in IBS. To ensure clarity and transparency, we have added a footnote below the table (lines 197 – 201) explaining that the indications of tolerability and scientific support should be interpreted as qualitative summaries rather than standardized quantitative evaluations.
Comments 10: Some of the data regarding gut microbiota is general – there is no specific information regarding changes in Bifidobacterium, Akkermansia, Faecalibacterium, etc.
Response 10: Thank you for the valuable comment. We fully understand the importance of a detailed analysis of specific changes in the gut microbiota, including genera such as Bifidobacterium, Akkermansia, and Faecalibacterium. However, this review was designed with the primary aim of critically analyzing the strengths and, above all, the limitations of the main dietary strategies used in patients with IBS. While a detailed and systematic description of microbial alterations is indeed relevant, including it would have considerably broadened the scope of the work, moving away from the main focus of the review.
Comments 11: On page 20, the authors mention the risk of orthorexia – it would be helpful to provide specific numerical data (e.g., screening rates).
Response 11: Thank you for the valuable comment. We have integrated quantitative data (lines 806 – 813), based on the study by Sultan et al. (Neurogastroenterol Motil, 2024). This study involved 202 IBS patients and 109 controls, finding that 33% of IBS patients scored ≥ 2 on the Sick, Control, One stone, Fat, Food (SCOF) questionnaire (indicative of eating disorders), compared to 16% in the control group (p < 0.001). Additionally, mean scores on the Eating Habits Questionnaire (EHQ) were significantly higher in the IBS group (82.9 ± 18.1 vs. 73.5 ± 16.9; p < 0.001), confirming a greater presence of orthorexic symptoms. These data enrich the discussion on the nutritional and psychological risks associated with restrictive diets in IBS patients.
Comments 12: Please supplement the citations in the section on probiotics (p. 22) with meta-analyses.
Response 12: Thank you for the valuable comment. In response to your suggestion, we have supplemented the section 2.5 (Dietary Supplements in IBS: Probiotics) with evidence from a recent meta-analysis by Ford et al. (2018). This meta-analysis, based on multiple randomized controlled trials, confirmed the efficacy of probiotics in reducing global IBS symptoms compared to placebo, although variability was observed depending on the strain and formulation. The text has been updated accordingly to reflect this addition (lines 935 – 939), enhancing the scientific rigor of the section.
Round 2
Reviewer 3 Report
Comments and Suggestions for Authors
Accept in present form